# The Primacy of Magnitude in Low-Rank Adaptation

**Zicheng Zhang**[1]*    **Haoran Li**[2]    **Yifeng Zhang**[1]    **Guoqiang Gong**[1]    **Jiaxing Wang**[1]

**Junxing Hu**[1]    **Pengzhang Liu**[1]    **Qixia Jiang**[1]

[1]`JD.com`
[2]`University of Chinese Academy of Sciences`

## Abstract

Low-Rank Adaptation (LoRA) offers a parameter-efficient paradigm for tuning large models. While recent spectral initialization methods improve convergence and performance over the naive "Noise & Zeros" scheme, their extra computational and storage overhead undermines efficiency. In this paper, we establish update magnitude as the fundamental driver of LoRA performance and propose LoRAM, a magnitude-driven "Basis & Basis" initialization scheme that matches spectral methods without their inefficiencies[1]. Our key contributions are threefold: *(i) Magnitude of weight updates determines convergence.* We prove low-rank structures intrinsically bound update magnitudes, unifying hyperparameter tuning in learning rate, scaling factor, and initialization as mechanisms to optimize magnitude regulation. *(ii) Spectral initialization succeeds via magnitude amplification.* We demystify that the presumed knowledge-driven benefit of the spectral component essentially arises from the boost in the weight update magnitude. *(iii) A novel and compact initialization strategy, LoRAM, scales deterministic orthogonal bases using pretrained weight magnitudes to simulate spectral gains.* Extensive experiments show that LoRAM serves as a strong baseline, retaining the full efficiency of LoRA while matching or outperforming spectral initialization across benchmarks.

## 1 Introduction

The rise of large pretrained models [1, 2, 3, 4, 5] has driven urgent needs for parameter-efficient fine-tuning (PEFT) methods [6, 7, 8, 9, 10, 11]. Among these, Low-Rank Adaptation (LoRA) [7] stands out for its *efficiency, flexibility, and stability*. By freezing pretrained weights and injecting trainable low-rank matrices, LoRA enables the update of less than 1% of the parameters, significantly reducing memory and compute costs. Its plug-and-play nature achieves easy integration into diverse models, facilitating model sharing and federated learning [12, 13]. Additionally, LoRA helps prevent catastrophic forgetting [14], making it well-suited for continual learning. These advantages have led to its wide adoption in multilingual NLP [15, 16, 17, 18] and multimodal applications [19, 20, 21, 22].

Despite achieved efficiency, the low-rank reparameterization constrains practical performance and convergence [14]. Besides the well-known "representation bottleneck" [23, 10, 24, 25, 26, 27, 28, 29], LoRA is highly sensitive to hyperparameters due to the non-convex and non-smooth loss landscape [30]. Effective training relies on careful tuning of rank [31, 32, 33], scaling factor [34], learning rate [35], initialization strategies [36, 37, 30], and preconditioning [30, 38, 39]. Recent works increasingly leverage information from pretrained weights [40, 41, 42] or task-specific data [43, 44, 39] to improve the "Noise & Zeros" baseline. Among these, PiSSA [40] pioneers the use of Singular Value Decomposition (SVD) for LoRA initialization, employing spectral components of pretrained

---

*Corresponding author: zhangzicheng6@jd.com
[1]Code is available here

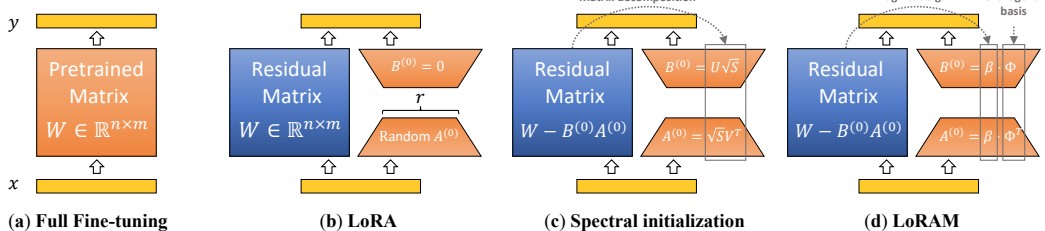

**(a) Full Fine-tuning**  **(b) LoRA**  **(c) Spectral initialization**  **(d) LoRAM**

Figure 1: We propose LoRAM, a magnitude-driven initialization method that enhances both the convergence and performance of LoRA while maintaining its efficiency. *Unlike spectral initialization, which precomputes and stores singular components $(U, V, S)$ [40], LoRAM uses deterministic orthogonal bases and derives scaling from pretrained weight statistics.* This elegant simplification is grounded in our analysis of LoRA through a novel lens of magnitude dynamics, where we show that the benefits of spectral values in scaling weight update magnitude can be effectively approximated.

weights to significantly enhance convergence and performance. Subsequent works [41, 44, 42, 39, 43] extend to diverse matrix decompositions, fostering a wave of knowledge-driven initialization schemes.

While spectral initialization [40, 41, 44, 39, 43] showcases considerable promise in convergence and performance, two fundamental challenges persist. First, they introduce **complexity by requiring additional matrix decomposition and storage overhead**, undermining usage efficiency in resource-constrained settings [45, 25, 46] and hindering seamless integration with deep learning libraries [6, 47]. Second, **their success remains poorly understood**. The common justification [40, 41, 43, 48] that spectral components preserve features better than random alternatives lacks theoretical grounding. Although recent works [44, 39] suggest that some specific initialization may approximate full-parameter gradients, the non-convex nature of LoRA renders training dynamics unpredictable.

In this paper, we demystify the knowledge-driven intuition behind spectral initialization and reveal that its effectiveness primarily stems from the magnitude scaling of weight updates. We design a minimal "Basis & Basis" initialization strategy, which demonstrates comparable performance without the overhead of SVD operations. Specifically, our key contributions can be summarized as:

• We identify **weight update magnitude as a fundamental principle** for analyzing and improving LoRA's training dynamics. This principle unifies previously independent factors, such as learning rate, scaling, and initialization, revealing their shared ability to control weight update strength and achieve comparable amplification effects when properly configured.

• We demonstrate that **initialization scheme critically shapes LoRA's weight magnitude dynamics**. Theoretically, we prove that LoRA naturally produces smaller updates than full fine-tuning, which limits its convergence and expressiveness. Moreover, we show that spectral initialization amplifies updates, providing a principled explanation for its effectiveness beyond knowledge-driven intuition.

• Guided by the magnitude principle, we propose **Magnitude-driven Initialization (LoRAM)** to make LoRA initialization efficient again. LoRAM employs a logarithmic magnitude factor to retain the benefits of spectral scaling, while directly scaling deterministic orthogonal bases to eliminate the need for decomposition and storage. Extensive experiments on language and vision-language tasks establish LoRAM as a strong and practical baseline, surpassing prior initialization schemes.

## 2 Magnitude Principle for Characterizing LoRA Dynamics

### 2.1 Preliminaries and Notations

Given a pretrained weight matrix $W \in \mathbb{R}^{n \times m}$, LoRA [7] reparameterizes the forward pass as

$$y = Wx + W_{\text{LoRA}}x = Wx + \alpha(BA - B^{(0)}A^{(0)})x, \tag{1}$$

where $B \in \mathbb{R}^{n \times r}$, $A \in \mathbb{R}^{r \times m}$ are trainable low-rank matrices with $r \ll \min(n, m)$, and $\alpha$ scales the update magnitude. The initialization term $B^{(0)}A^{(0)}$ can be absorbed into $W$ for the convenience, as illustrated in Figure 1(b). Given a loss function $L$, LoRA updates are computed as:

$$\nabla_A L = \alpha B^{\top} \frac{\partial L}{\partial y} x^{\top} = \alpha B^{\top}(\nabla_W L), \quad \nabla_B L = \alpha \frac{\partial L}{\partial y} x^{\top} A^{\top} = \alpha(\nabla_W L)A^{\top}. \tag{2}$$

**Magnitude metric.** To elucidate how LoRA affects the training process, we analyze the dynamics of the parameters $A$, $B$, and the resulting weight update $W_{\text{LoRA}}$. Specifically, we define the weight

magnitude as $\nu[W_{\text{LoRA}}] = \frac{1}{mn}\|W_{\text{LoRA}}\|_F^2$, which serves as a central metric in our study. Assuming independent and zero-mean entries [49], the expected weight magnitude is given by $\mathbb{E}[\nu[BA]] = r\,\mathbb{E}[\nu[B]]\,\mathbb{E}[\nu[A]]$. In the asymptotic regime where $m$ and $n$ are large, $\nu[W_{\text{LoRA}}]$ is approximated with the variance of $W_{\text{LoRA}}$, and $\nu[BA] \approx r\,\nu[B]\,\nu[A]$. Our analysis is motivated by the fact that LoRA introduces no change to the pretrained weights initially, *i.e.*, $\nu[W_{\text{LoRA}}^{(0)}] = 0$, while its effect emerges gradually through training. Therefore, we use the term "magnitude" instead of "variance" to highlight the cumulative growth of $W_{\text{LoRA}}$. In Appendix B, we also present a theoretical insight showing that parameter magnitude is a key determinant of LoRA's expressiveness.

## 2.2 Effect of Hyperparameters on Update Magnitude

Let $\Delta W_{\text{LoRA}}^{(t)}$ denote the weight update at step $t$ for LoRA framework, and $W_{\text{LoRA}}^{(t)} = \sum_{i=0}^{t-1} \Delta W_{\text{LoRA}}^{(i)}$ represent cumulative adaptation. Given a learning rate $\eta$, we expand $\Delta W_{\text{LoRA}}^{(t)}$ using gradient update rules, leading to the following formulations for the update magnitude[2]:

$$\Delta W_{\text{LoRA}}^{(t)} = \alpha\eta\left(B^{(t)}\nabla_A L^{(t)} + \nabla_B L^{(t)} A^{(t)} + \eta\nabla_B L^{(t)}\nabla_A L^{(t)}\right), \qquad (3)$$

$$\text{and}\quad \nu[\Delta W_{\text{LoRA}}^{(t)}] \approx r\alpha^2\eta^2\left(\nu[B^{(t)}]\nu[\nabla_A L^{(t)}] + \nu[\nabla_B L^{(t)}]\nu[A^{(t)}]\right). \qquad (4)$$

These equations indicate the complex interplay of multiple hyperparameters, distinct from the full-parameter updates given by $\Delta W^{(t)} = -\eta\nabla_W L^{(t)}$. We investigate the interplay among the learning rate $\eta$, scaling factor $\alpha$, and initialization magnitude, revealing a quantifiable equivalence relationship.

**Proposition 1** (Parameter Scaling Equivalence). *For LoRA layers defined in Eq.* (1)*, consider decomposing the scaling factor* $\alpha = \alpha'\alpha_A\alpha_B$*, where* $\alpha', \alpha_A, \alpha_B \in \mathbb{R}^+$*. Under the commonly used optimization frameworks with negligible numerical errors, the following parametrization schemes exhibit dynamical equivalence throughout training: For all iterations* $t \geq 0$*,* $\Delta W_{\text{LoRA}}^{(t)} = \Delta\tilde{W}_{\text{LoRA}}^{(t)}$ *and* $W_{\text{LoRA}}^{(t)} = \tilde{W}_{\text{LoRA}}^{(t)}$*, where* $\tilde{A}^{(t)}$*,* $\tilde{B}^{(t)}$ *and* $\tilde{W}^{(t)}$ *represent the re-parameterized versions.*

| | *Original* | *SGD* | *Adam* |
|---|---|---|---|
| *Representation* | $\alpha BAx$ | $\alpha'\tilde{B}\tilde{A}x$ | $\alpha'\tilde{B}\tilde{A}x$ |
| *Initialization* | $A^{(0)} = A_{init}, B^{(0)} = B_{init}$ | $\tilde{A}^{(0)} = \alpha_A A_{init}, \tilde{B}^{(0)} = \alpha_B B_{init}$ | $\tilde{A}^{(0)} = \alpha_A A_{init}, \tilde{B}^{(0)} = \alpha_B B_{init}$ |
| *Learning Rates* | $\eta_A > 0, \eta_B > 0$ | $\eta_{\tilde{A}} = \alpha_A^2\eta_A, \eta_{\tilde{B}} = \alpha_B^2\eta_B$ | $\eta_{\tilde{A}} = \alpha_A\eta_A, \eta_{\tilde{B}} = \alpha_B\eta_B$ |

**Remarks.** This equivalence underscores how hyperparameters collectively regulate update magnitude, effectively reducing the search space for optimal configurations. A striking implication is that, increasing $\eta_B$ in LoRA+ [35] is identical to scaling $\alpha$ in RsLoRA [34] under the "Noise & Zeros" initialization, highlighting the critical role of initialization magnitude in shaping LoRA's training dynamics, which is rarely discussed in prior works. For the non-zero initialization, it is advisable to first adjust the initialization magnitudes and learning rates for moderate improvements, as modifying $\alpha$ inherently combines the effects of both and may result in drastic and unpredictable changes.

To demonstrate the joint effect of hyperparameters on LoRA dynamics, we conduct a controlled experiment using a 5-layer MLP with "Noise & Zeros" initialized LoRA layers, setting the intermediate dimension to 400 and the LoRA rank to 25. The network is trained on synthetic data under various hyperparameter settings, consequently using SGD and Adam optimizers with $\eta = 5 \times 10^{-5}$. As shown in Figure 2(a), all settings with $\alpha = 16$ result in identical loss trajectories and parameter evolution, confirming the theoretical predictions. In contrast, weight updates deviate significantly when using $\alpha = 1$ with $\eta$ scaled by 4, indicating that equivalence holds only under specific rules.

## 2.3 Magnitude Limitation Rooted in Low-Rank Structure

Guided by the established equivalence framework, we fix $\alpha = 1$ to eliminate the interference of scaling factors, which is the most commonly-used configuration in practical implementation. In the following, we prove initialization magnitudes and other factors critically influence the weight update.

**Proposition 2** (Parameter Magnitude Dynamics). *Consider LoRA parameters updated with the same learning rate* $\eta$*. Assume:* $A^{(0)} \sim \mathcal{N}(\mathbf{0}, \sigma_A^2 I)$*,* $B^{(0)} \sim \mathcal{N}(\mathbf{0}, \sigma_B^2 I)$*,* $\nabla_W L^{(t)} \sim \mathcal{N}(\mathbf{0}, \sigma_L^2 I)$*, and*

---

[2]See Appendix for derivation and proof

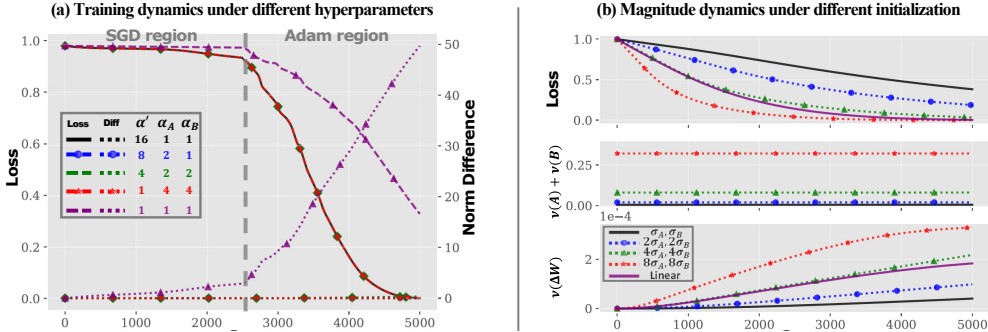

Figure 2: (a) Validation of Proposition 1. Each curve represents a model with unique hyperparameters. The norm difference (right axis) aggregates Frobenius norm discrepancies between the baseline model (black) and others across layers. Purple and other curves share identical learning rates but diverge due to differing initialization magnitudes. Equivalent optimization trajectories emerge from diverse hyperparameter combinations under both SGD and Adam optimizers. (b) Validation of Proposition 2. The black curve represents random orthogonal initialization. Parameter magnitudes are predominantly governed by initialization scaling, resulting in smaller weight changes compared to conventional linear layers. This necessitates the magnitude scaling in enhancing LoRA performance.

$\mathbb{E}[\langle A^{(t)}, \nabla_A L^{(t)} \rangle] = \mathbb{E}[\langle B^{(t)}, \nabla_B L^{(t)} \rangle] = 0$. *Under these conditions, the parameter magnitudes* $\boldsymbol{\nu}_t = \begin{bmatrix} \mathbb{E}[\nu[A^{(t)}]], \mathbb{E}[\nu[B^{(t)}]] \end{bmatrix}^T$ *evolve as a linear dynamical system. Its exponential solution admits the linearized approximation under small-$\eta$ regime:*

$$\boldsymbol{\nu}_t = \left( I + \begin{bmatrix} 0 & \gamma_B \\ \gamma_A & 0 \end{bmatrix} \right) \boldsymbol{\nu}_{t-1} \approx \begin{bmatrix} \sigma_A^2 + t\gamma_B\sigma_B^2 \\ \sigma_B^2 + t\gamma_A\sigma_A^2 \end{bmatrix}, \tag{5}$$

*where* $\gamma_A = m\eta^2\sigma_L^2$, $\gamma_B = n\eta^2\sigma_L^2$. *This further yields the evolution of weight update magnitude:*

$$\nu[W_{LoRA}^{(t)}] \approx k_1\gamma t + \mathcal{O}(\gamma^2 t^2), \text{ where } \gamma = \eta^2\sigma_L^2, \ k_1 = r(m\sigma_A^4 + n\sigma_B^4). \tag{6}$$

**Remarks.** This analysis uncovers essential properties of LoRA initialization and optimization trajectory. First, since $\gamma_A$ and $\gamma_B$ are very small values ($\ll 1$), the magnitudes of parameters $A^{(t)}$ and $B^{(t)}$ remains nearly unchanged throughout training, potentially constraining the representation capacity of the learned model. Moreover, unlike full-parameter tuning, which evolves at a linear rate of $\gamma$, *the low-rank structure introduces a proportional factor $k_1$, significantly slowing updates*. For instance, the naive "Noise & Zeros" initialization yields $k_1 = \frac{r}{m}$, while the dimension $m$ in large models like LLaMA [3] is in the thousands or more. Despite the quadratic term accelerates growth, small gradients may temper this effect in the later training stages.

Figure 2(b) visualizes LoRA magnitude dynamics during training. We use the same network as in Figure 2(a) but apply a nonzero initialization with $\sigma_A = \sigma_B = \frac{1}{20}$. We explore a regular MLP with the same magnitude denoted as "Linear", and a group of networks with larger initialization magnitudes. Notably, while the loss decreases significantly, the magnitudes of $A$ and $B$ remain nearly unchanged throughout training. The magnitude evolution of $W$ reveals that the basic LoRA with theoretically $k_1 = \frac{1}{16}$, grows substantially slower than the regular MLP. Increasing the initialization magnitude effectively accelerates the growth of $W$, aligning with our theoretical analysis.

Integrating analyses in this section, we derive a **magnitude principle** for LoRA development:

> *A valid improvement to LoRA convergence will enhance weight update magnitude* $\nu[W_{LoRA}]$.

As shown in Proposition 2, magnitude principle could unify and explain improvement factors in existing works, including the learning rate, scaling factor, gradients, rank and initialization schemes.

## 3 Demystifying Spectral Gains with Magnitude Principle

The sheer scale of modern neural networks complicates the determination of optimal LoRA initialization across layers. Drawing inspiration from recent spectral initialization methods [40, 41, 43, 44, 39], which have demonstrated improved convergence and task performance, we reinterpret their effectiveness through the lens of the magnitude principle and introduce a magnitude-driven initialization

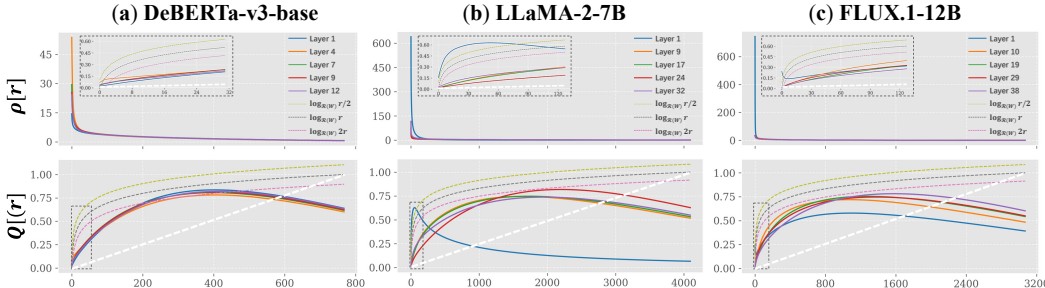

Figure 3: Illustration of spectral gain factor $Q[r]$ defined in Eq. (10) and spectral concentration factor $\rho[r]$ defined in Eq. (8) across DeBERTa-v3-base [50], LLaMA-2-7B [3] and FLUX.1-12B [51]. Values are computed from uniformly sampled layers. The white dotted line represents the linear growth rate of naive LoRA weight magnitudes, while spectral initialization exhibits faster growth. Due to its concave nature, we approximate the spectral gain factor using a logarithmic function.

method called LoRAM. Notably, these methods requires extra SVD computations and storage, leading to increased resource overhead and implementation complexity. In contrast, LoRAM mitigates these drawbacks and achieves even better performance. In the following, we take the seminal work PiSSA [40] as a representative baseline and ablate other methods in experiments (see Section 4.3).

## 3.1 Magnitude Gain in Spectral Initialization

The PiSSA method [40] initializes LoRA using the spectral decomposition of pretrained weight matrix $W = USV^\top$, which has a rank of $\mathcal{R}[W]$. The spectral initialization is defined as:

$$A^{(0)} = A_{\text{SVD}} = \sqrt{S_r}V_{:,:r}^\top, \quad B^{(0)} = B_{\text{SVD}} = U_{:,:r}\sqrt{S_r}, \tag{7}$$

where $S_r \in \mathbb{R}^{r \times r}$ contains the top-$r$ singular values, and $U \in \mathbb{R}^{n \times n}$, $V \in \mathbb{R}^{m \times m}$ are the left and right singular vector matrices. While prior works [40, 41, 43, 48] attribute PiSSA's success to its ability to preserve principal components, we show that its key advantage lies in singular value weighting. By redistributing dominant variance into the initialization, PiSSA facilitates adaptive magnitude updates across layers, accelerating convergence.

Consider the statistics of the top-$r$ singular values, we define the spectral concentration factor as:

$$\rho[r] \triangleq \frac{\mathbb{E}_r[s]^2}{\mathbb{E}_{\mathcal{R}[W]}[s^2]} = \frac{\left(\frac{1}{r}\sum_{i=1}^r s_i\right)^2}{\frac{1}{\mathcal{R}[W]}\sum_{i=1}^{\mathcal{R}[W]} s_i^2}. \tag{8}$$

This factor captures the concentration of energy in the top-$r$ singular values. We then reformulate

$$\nu(A_{\text{SVD}}) = \mathbb{E}_r[s]\nu[V_{:,:r}] = \sqrt{\frac{n\rho[r]\nu[W]}{m\mathcal{R}[W]}}, \ \nu(B_{\text{SVD}}) = \mathbb{E}_r[s]\nu[U_{:,:r}] = \sqrt{\frac{m\rho[r]\nu[W]}{n\mathcal{R}[W]}}. \tag{9}$$

Essentially, $\rho[r]$ acts as a scaling factor that redistributes variance from the pretrained weight matrix, influencing the magnitude of updates during training. Since $\rho[r]$ monotonically decreases with $r$, its impact is more pronounced for smaller $r$, making it particularly relevant for LoRA applications.

**Spectral Gain Factor.** Taking the above magnitudes into the dynamics in Eq. (6) further derives:

$$k_1 = Q[r](m+n)\nu[W], \quad 0 \le Q[r] \triangleq \frac{\rho[r]r}{\mathcal{R}[W]} \le 1. \tag{10}$$

Given that $\nu[W] \sim \mathcal{O}(\min(\frac{1}{m}, \frac{1}{n}))$, this results in a gain factor of at least $Q[r]$, which we term as the "spectral gain factor". As shown in Figure 3, the spectral gain factor $Q[r]$ exhibits bounded variation in $[0, 1]$ with characteristic concavity, which can be formally derived via Jensen's inequality. Although SVD components are not completely independent preventing the theoretical monotonic increase in $Q[r]$, this concavity also suggests that the gain effect is more pronounced when $r$ is small, reinforcing the effectiveness of spectral initialization for LoRA in the parameter-efficient manner.

## 3.2 Efficient Magnitude-driven Initialization with LoRAM

We propose LoRAM to achieve similar magnitude update rate in Eq. (10) like PiSSA while eliminating spectral computation. As depicted in Algorithm 1, LoRAM initializes the parameter matrices as:

$$A^{(0)} = A_{\text{LoRAM}} = \beta \cdot \Phi_m^\top, \quad B^{(0)} = B_{\text{LoRAM}} = \beta \cdot \Phi_n, \quad \beta = \left(\frac{Q[r]\cdot\nu[W]}{\nu[\Phi_n\Phi_m^\top]}\right)^{\frac{1}{4}}. \tag{11}$$

---

**Algorithm 1** LoRAM Initialization Procedure

---

**Input:** Pretrained weight $W \in \mathbb{R}^{n \times m}$, target rank $r$
**Output:** Initialized parameters $A^{(0)}, B^{(0)}, W$

---

$\Phi_n, \Phi_m \leftarrow \text{get\_basis(n, r), get\_basis(m, r)}$          ▷ Generate basis matrices, *e.g.*, Eq (12)

$\beta \leftarrow \left( \frac{\log_{\min(n,m)}(r) \cdot \nu[W]}{\nu[\Phi_n \Phi_m^\top]} \right)^{1/4}$          ▷ Compute magnitude gain factor

$B^{(0)}, A^{(0)}, W \leftarrow \beta \cdot \Phi_n, \beta \cdot \Phi_m^\top, W - \beta^2 \cdot \Phi_n \Phi_m^\top$          ▷ Initialize parameters

---

Here $\Phi_n$ and $\Phi_m$ denote the first $r$ columns of an $n$- and $m$-dimensional orthogonal basis matrices, respectively. Given that $\nu[\Phi_n] = \frac{1}{n}$ and $\nu[\Phi_m] = \frac{1}{m}$, LoRAM achieves the similar magnitude as PiSSA with $k_1 = Q[r](m+n)\nu[W]$. We can also derive $\nu[B^{(0)} A^{(0)}] = Q[r]\nu[W]$, implying LoRAM inherently ensures numerical stability and moderate corrections to the pretrained weight.

**Logarithmic Gain Factor.** Due to the concave nature of $Q[r]$, we approximate its analytical form using an asymptotic expansion: $Q[r] \approx \log_{\min(n,m)}(r)$. As illustrated in Figure 3, this logarithmic function effectively captures the monotonic increase nature, providing a predictable improvement than LoRA and PiSSA particularly when using a small rank $r$.

**Deterministic Basis Matrix.** To eliminate the need for storing initialization buffers, we adopt an analytic approach instead of random initialization. Specifically, we employ the Discrete Sine Transform (DST) basis due to its simplistic mathematical definition:

$$\Phi_m[i,j] = \sqrt{\frac{2}{m+1}} \sin\left( \frac{(i+1)(j+1)\pi}{m+1} \right), \quad 0 \le i, j < m. \tag{12}$$

This formulation constructs orthogonal matrices of arbitrary dimensions, ensuring reproducibility across different devices while providing provable statistical properties: $\mathbb{E}[\Phi_m] = 0$ and $\nu[\Phi_m] = \frac{1}{m}$. One may wonder if randomness is required for initialization, we find it unnecessary in LoRA. In fact, DST even slightly outperforms random strategies in our ablation experiments (see Section 4.3).

**Efficiency and Compatibility.** Since $\beta$ and $\Phi$ avoid complex matrix operations, LoRAM retains the efficiency and storage footprint of naive LoRA. As it only modifies initialization, LoRAM remains plug-and-play, integrating seamlessly into any pipeline that supports standard LoRA. This is especially valuable for modern large models built on fixed and highly optimized frameworks. In contrast, other initialization methods require costly preprocessing, such as matrix generation [44, 43] or decomposition [40, 42, 41], which complicates adoption in standard workflows.

## 4 Experiments

We conduct comprehensive experiments to evaluate LoRAM efficiently implemented via the PEFT library [6]. Following conventional settings [40, 41, 35], we assess performance on language tasks and extend the evaluation to vision-language tasks, demonstrating LoRAM's generalization across diverse models and modalities. All experiments are run on servers with 8 NVIDIA H800 GPUs.

**Baselines.** While extensive research on LoRA has explored aspects like structural modifications and rank control, these directions are largely orthogonal to our focus on hyperparameter analysis within the naive LoRA framework. In line with this, we compare LoRAM with the naive LoRA (ICLR 2022) [7], as well as several representative hyperparameter tuning strategies. We first consider weight-driven initialization (marked "§"), including PiSSA (NeurIPS 2024) [40], which uses the top-$r$ singular vectors and values of pre-trained weights; MiLoRA (NAACL 2025) [41], which utilizes the last $r$ singular vectors and values; and OLoRA [42], which applies orthogonal initialization via QR decomposition. All these methods adopt a fixed scaling factor $\alpha = 1$. We then include RsLoRA [34] (marked "†"), which enhances performance by setting $\alpha = \sqrt{r}$, and LoRA+ (ICML 2024) [35] (marked "‡"), which increases the learning rate with the recommended $\eta_B = 4\eta_A$. We also evaluate data-driven initialization in the ablation study, including LoRA-GA (NeurIPS 2024) [52] and CorDA (NeurIPS 2024) [43], which require extra pipeline to leverage training data information. Most of these baselines have been integrated and validated in the PEFT library.

Table 1: Comparison of LoRAM versus hyperparameter tuning baselines on NLG tasks. Experiments conducted with LLaMA2-7B model using two ranks, reporting mean ± std results (%) over three runs. Bold and underlined values represent the best and second-best performances, respectively.

| Rank | #Param | Method | GSM8K | MATH | HumanEval | MBPP | Commonsense |
|------|--------|--------|-------|------|-----------|------|-------------|
| N/A | 6738M | Full FT | $60.34 \pm 1.32$ | $11.74 \pm 0.63$ | $32.30 \pm 1.26$ | $39.27 \pm 1.01$ | $79.20 \pm 1.20$ |
| 16 | 40M | LoRA | $31.51 \pm 0.31$ | $4.16 \pm 0.27$ | $15.98 \pm 0.20$ | $\underline{28.65} \pm 0.47$ | $66.56 \pm 1.21$ |
| | | RsLoRA[†] | $\underline{39.04} \pm 0.53$ | $4.94 \pm 0.40$ | $\underline{18.85} \pm 0.66$ | $28.10 \pm 0.64$ | $73.24 \pm 0.84$ |
| | | LoRA+[‡] | $31.69 \pm 0.64$ | $3.98 \pm 0.38$ | $18.54 \pm 0.52$ | $28.00 \pm 0.81$ | $72.19 \pm 1.43$ |
| | | MiLoRA[§] | $29.70 \pm 0.42$ | $4.18 \pm 0.21$ | $14.69 \pm 0.66$ | $27.23 \pm 0.53$ | $67.90 \pm 1.20$ |
| | | OLoRA[§] | $35.83 \pm 0.58$ | $4.80 \pm 0.53$ | $16.58 \pm 0.38$ | $27.44 \pm 0.76$ | $73.48 \pm 1.09$ |
| | | PiSSA[§] | $37.68 \pm 0.45$ | $\underline{5.16} \pm 0.41$ | $18.37 \pm 0.49$ | $28.62 \pm 0.68$ | $\underline{73.72} \pm 1.05$ |
| | | LoRAM[§] | $\mathbf{40.32} \pm 0.43$ | $\mathbf{5.30} \pm 0.37$ | $\mathbf{18.92} \pm 0.55$ | $\mathbf{28.83} \pm 0.63$ | $\mathbf{75.19} \pm 1.10$ |
| 128 | 320M | LoRA | $40.27 \pm 0.70$ | $4.72 \pm 0.43$ | $20.11 \pm 0.32$ | $28.84 \pm 0.37$ | $73.64 \pm 1.13$ |
| | | RsLoRA[†] | $50.38 \pm 0.37$ | $\mathbf{7.32} \pm 0.28$ | $21.32 \pm 0.70$ | $30.73 \pm 0.43$ | $77.01 \pm 1.17$ |
| | | LoRA+[‡] | $40.41 \pm 0.67$ | $5.28 \pm 0.53$ | $20.71 \pm 0.88$ | $29.13 \pm 0.78$ | $\underline{78.19} \pm 1.33$ |
| | | MiLoRA[§] | $39.81 \pm 0.89$ | $5.18 \pm 0.58$ | $20.39 \pm 0.21$ | $29.95 \pm 1.05$ | $74.29 \pm 1.09$ |
| | | OLoRA[§] | $50.10 \pm 0.64$ | $7.01 \pm 0.56$ | $20.72 \pm 0.67$ | $30.21 \pm 0.89$ | $\mathbf{78.61} \pm 0.97$ |
| | | PiSSA[§] | $\mathbf{51.48} \pm 0.77$ | $7.04 \pm 0.54$ | $\underline{21.62} \pm 0.48$ | $\underline{31.07} \pm 0.68$ | $77.28 \pm 0.98$ |
| | | LoRAM[§] | $\underline{51.12} \pm 0.73$ | $\underline{7.25} \pm 0.68$ | $\mathbf{22.03} \pm 0.56$ | $\mathbf{31.53} \pm 0.72$ | $77.81 \pm 0.96$ |

Table 2: Comparison of LoRAM versus hyperparameter tuning baselines on GLUE benchmark. Experiments conducted with the DeBERTa-v3-base model using rank 8, reporting mean results over three runs. Bold and underlined values represent the best and second-best performances, respectively.

| Method | #Param | MNLI | SST-2 | MRPC | CoLA | QNLI | QQP | RTE | STS-B |
|--------|--------|------|-------|------|------|------|-----|-----|-------|
| Full FT | 184M | 88.31 | 93.57 | 89.46 | 67.26 | 92.80 | 91.52 | 83.75 | 86.87 |
| LoRA | 1.33M | 90.23 | $\mathbf{95.87}$ | 84.06 | 63.56 | 93.88 | 90.55 | 50.18 | 87.20 |
| RsLoRA[†] | 1.33M | 90.33 | 95.64 | 86.38 | 64.85 | 93.97 | 90.26 | 60.31 | 88.37 |
| LoRA+[‡] | 1.33M | $\underline{90.37}$ | 95.32 | 87.54 | 64.79 | $\mathbf{94.32}$ | 90.93 | 65.37 | $\underline{89.20}$ |
| MiLoRA[§] | 1.33M | 90.28 | $\underline{95.75}$ | 87.00 | 62.58 | 93.97 | 90.83 | 54.87 | 87.74 |
| OLoRA[§] | 1.33M | 90.19 | 94.83 | 88.72 | $\mathbf{65.59}$ | 93.36 | 90.74 | 74.09 | 88.52 |
| PiSSA[§] | 1.33M | $\mathbf{90.38}$ | 95.64 | $\underline{89.21}$ | 65.06 | 93.84 | $\underline{91.35}$ | $\underline{74.36}$ | 88.90 |
| LoRAM[§] | 1.33M | 90.34 | 95.29 | $\mathbf{89.95}$ | $\underline{65.53}$ | $\underline{94.08}$ | $\mathbf{91.70}$ | $\mathbf{74.72}$ | $\mathbf{89.93}$ |

## 4.1 Evaluating the Performance on Natural Language Tasks

**Nature Language Generation (NLG).** As shown in Table 1, we conduct supervised fine-tuning of LLaMA 2-7B [3] on math, coding, and commonsense reasoning tasks. Our setup strictly follows PiSSA [40], using the AdamW optimizer with a batch size of 128, a learning rate of $2 \times 10^{-5}$, a warmup ratio of 0.03, and no weight decay. All experiments are performed on subsets containing 100K data points for one epoch to minimize training overhead. For math tasks, the model is tuned on MetaMathQA [53] and evaluated on GSM8K [54] and MATH [53] validation sets. For coding tasks, we use CodeFeedback [55] as training dataset, with evaluations on HumanEval [56] and MBPP [57]. For commonsense tasks, model is tuned on Commonsense170K [58], and we report averaged accuracy on eight sub-datasets. The results in Table 1 demonstrate that LoRAM consistently outperforms LoRA variants across diverse tasks and rank settings, without requiring matrix decomposition.

**Nature Language Understanding (NLU).** We evaluate the NLU performance by fine-tuning the DeBERTa-v3-base model [50] with a rank of 8 on eight tasks in the GLUE benchmark [59]. We utilize scripts from the Transformers Library [47] to ensure a fair comparison. All methods are trained with a learning rate of $1 \times 10^{-4}$ for 3 training epochs, except for MRPC, which uses 5 epochs due to its smaller size. We report overall matched and mismatched accuracy on MNLI, Matthew's correlation on CoLA, Pearson correlation on STS-B, and accuracy on the other datasets. As shown in Table 2, LoRAM achieves competitive performance against PiSSA across most tasks.

## 4.2 Evaluating the Performance on Vision-Language Tasks

**Text-to-image synthesis.** We adapt the advanced FLUX.1-12B [51] to address the image customization task, implementing LoRA, PiSSA, and LoRAM under identical configurations: a learning rate of $1 \times 10^{-4}$, a batch size of 1 and 1,000 iterations. We set the rank to 8, optimizing 9.3 million

Table 3: Comparing LoRAM with other LoRA variants on LLaVA for multimodel tasks. Bold and underlined values indicate the top and second-best performances, respectively.

| Method | $\text{MME}_{\text{Cog}}$ | $\text{MME}_{\text{Per}}$ | MMMU | AI2D | ChartQA | OCRBench | TextVQA | ScienceQA |
|---|---|---|---|---|---|---|---|---|
| Full FT | 280 | 1541 | 0.355 | 0.583 | 0.251 | 0.361 | 0.597 | 0.722 |
| LoRA | 278 | 1402 | 0.331 | 0.557 | 0.231 | 0.333 | 0.536 | 0.684 |
| RsLoRA[†] | 274 | 1385 | 0.334 | **0.573** | 0.227 | 0.328 | 0.539 | 0.694 |
| LoRA+[‡] | 283 | 1389 | 0.341 | 0.565 | 0.229 | 0.331 | 0.545 | 0.690 |
| MiLoRA§ | 285 | 1354 | 0.340 | 0.564 | 0.220 | 0.335 | 0.536 | 0.681 |
| OLoRA§ | 288 | 1404 | 0.345 | 0.565 | 0.228 | 0.330 | 0.540 | 0.677 |
| PiSSA§ | **311** | **1411** | 0.344 | 0.564 | 0.232 | **0.338** | 0.547 | 0.686 |
| LoRAM§ | 308 | 1406 | **0.350** | 0.571 | **0.238** | 0.336 | **0.551** | **0.700** |

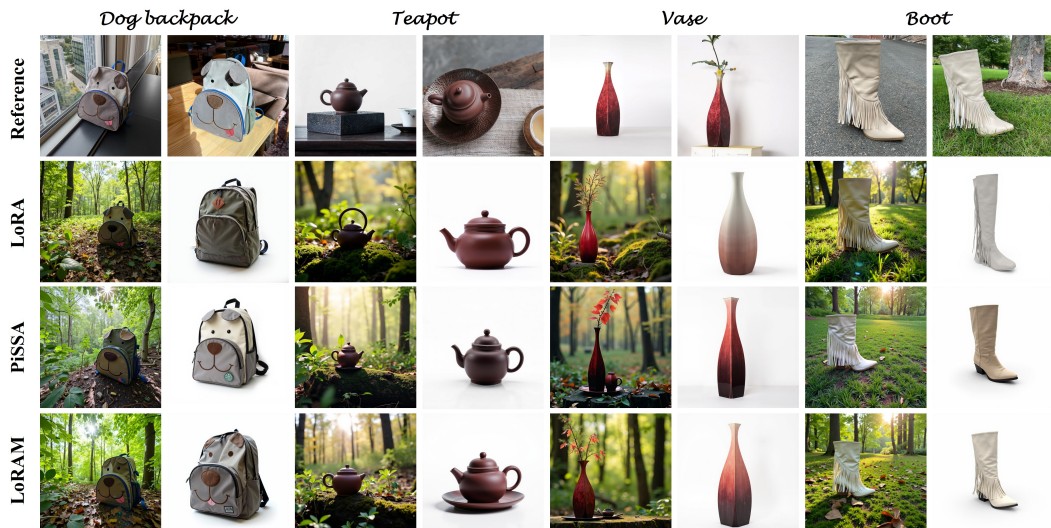

Figure 4: Comparison of LoRA, PiSSA, and LoRAM on image customization task. Experiments conducted with the state-of-the-art FLUX.1-12B model using rank 8.

parameters while maintaining computational efficiency on a single GPU. The training data and prompt template adhere to DreamBooth's protocol [22]. Qualitative results in Figure 4 demonstrate that LoRAM exhibits marginally superior performance in detail fidelity compared to PiSSA.

**Image-to-text generation.** Following the pipeline of LLaVA [60], we employ CLIP-ViT-L/14 [61] as the visual encoder, Vicuna-13B [62] as the text decoder, and a new visual resampler [63] as the connector. In the pre-training stage, we fine-tune only the perceiver resampler using the CC-595K dataset [60] for one epoch. During the subsequent instruction-tuning stage, we fine-tune both Vicuna and the resampler using a 656K mixture dataset [60]. The learning rate is set to $2 \times 10^{-5}$, and the batch size is 128. We follow the official implementation and use a rank of 64. As shown in Table 3, LoRAM achieves favorable performance across multiple multimodal benchmarks.

**Training curves.** We provide representative training loss curves of diverse initialization methods in Figure 5. It can be noticed that LoRAM is able to have a faster convergence rate in the early stages compared to other LoRA variants and incur smaller losses in the end.

## 4.3 Ablating the Magnitude Principle in LoRA improvements

We conduct ablation experiments on LLaMA-2-7B [3] under the NLG setting, focusing on the effect of magnitude gain factor, the choice of basis matrix, and the validation of the magnitude principle.

**Magnitude Gain Factor.** As shown in Table 4, we first evaluate different values of $Q[r]$ and observe that increasing its value slightly improves performance. We further introduce a "tracking mode" that adjusts $\beta$ in Algorithm 1 based on the initialization magnitudes from reference methods. Specifically, we set $\beta = \sqrt{\frac{\mathbb{V}[B_{\text{ref}} A_{\text{ref}}]}{\mathbb{V}[\Phi_n \Phi_m^\top]}}$, where $B_{\text{ref}}$ and $A_{\text{ref}}$ are initialization matrix using weight-driven [40, 41] or data-driven approaches [43, 44]. Under this mode, LoRAM essentially matches the performance of prior methods, confirming that magnitudes govern its performance. We also observe that PiSSA,

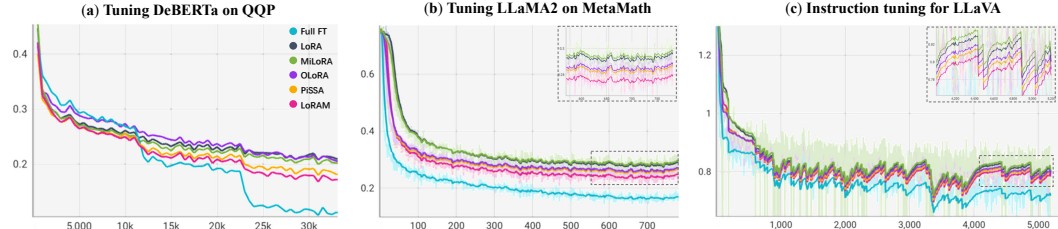

Figure 5: Illustration of training loss curves. LoRAM achieves comparative convergence dynamics to PiSSA across diverse models and benchmarks. See tables and texts for the evaluation results.

Table 4: Results of ablation study. "Orth." denotes the random orthogonal initialization. The left and right sides of the slash indicate the results of the method and the tracking mode, respectively. The average value is calculated over all the ranks and tasks to compare the overall trend of change.

| Rank | Task | Q[r] | | | Basis | | Weight-driven | | Data-driven | | + RsLoRA | |
|---|---|---|---|---|---|---|---|---|---|---|---|---|
| | | $\log \frac{r}{2}$ | $\log r^*$ | $\log 2r$ | Orth. | Gaussian | MiLoRA | PiSSA | CorDA | LoRA-GA | LoRA-GA | LoRAM |
| 16 | MATH | 5.08 | 5.30 | 5.18 | 4.74 | 4.62 | 4.18 / 4.05 | 5.16 / 5.10 | 4.60 / 4.14 | 5.73 / 3.76 | 7.94 | 7.22 |
| | GSM8k | 40.1 | 40.3 | 40.7 | 36.3 | 35.8 | 29.7 / 29.6 | 37.6 / 36.7 | 36.2 / 30.7 | 45.7 / 30.9 | 51.5 | 52.1 |
| | MBPP | 28.8 | 28.8 | 28.3 | 28.6 | 27.5 | 27.2 / 27.8 | 28.6 / 29.1 | 25.7 / 28.3 | 28.3 / 27.8 | 33.9 | 31.5 |
| | HumanEval | 17.1 | 18.9 | 17.1 | 17.7 | 17.7 | 14.6 / 14.7 | 18.3 / 17.3 | 15.2 / 15.2 | 19.5 / 17.7 | 22.0 | 18.3 |
| 128 | MATH | 7.52 | 7.25 | 7.51 | 7.40 | 7.62 | 5.18 / 5.04 | 7.04 / 6.95 | 6.26 / 5.04 | 9.18 / 7.32 | 9.08 | 11.1 |
| | GSM8k | 50.7 | 51.1 | 50.4 | 50.2 | 49.8 | 39.8 / 39.2 | 51.4 / 49.5 | 44.5 / 40.3 | 54.4 / 50.8 | 53.6 | 59.4 |
| | MBPP | 31.5 | 31.5 | 32.3 | 31.3 | 32.8 | 29.9 / 30.2 | 31.0 / 29.6 | 29.6 / 29.9 | 32.0 / 30.2 | 31.7 | 38.1 |
| | HumanEval | 20.7 | 22.0 | 22.6 | 23.2 | 22.6 | 20.3 / 19.5 | 21.6 / 20.3 | 20.1 / 18.9 | 24.4 / 20.7 | 26.8 | 31.7 |
| **Average value** | | 25.1 | 25.5 | 25.6 | 24.9 | 24.8 | 21.1 / 21.2 | 25.0 / 24.3 | 22.7 / 21.6 | 27.4 / 23.6 | 29.5 | 31.1 |

which selects the top-$r$ singular values, outperforms MiLoRA, which selects the last $r$ singular values. This indicates that leveraging the large dominant singular values enhances performance.

**Basis Matrix.** We find that the choice of basis matrix generally has limited impact. For instance, replacing the DST basis with a random orthogonal or Gaussian matrix just results in only minor performance degradation. In tracking mode, substituting the SVD-derived basis with DST does not significantly impact performance. A notable exception is LoRA-GA [44], which approximates the full-parameter gradient at initialization. Nonetheless, we emphasize that tracking mode fails not due to incorrectness of magnitude principle, but because the LoRA-GA matrix form maximizes gradient magnitudes of Eq. (2), making it irreplaceable by alternatives and validating magnitude principle[3].

**Upper Bound of Magnitude Scaling.** While increasing update magnitude generally improves performance, the benefit is not unlimited. For example, applying RsLoRA to LoRA-GA yields a clear gain at rank 8, but the improvement diminishes and may even reverse at rank 128. This suggests that magnitude scaling should be applied conservatively at higher ranks, since larger ranks inherently amplify updates, as demonstrated in our Proposition 2. Given that data-driven methods involve costly gradient and SVD computations, we recommend LoRAM with RsLoRA as a more efficient and scalable alternative for accelerating LoRA convergence and performance.

## 5 Conclusion

In this paper, we explore the magnitude principle of Low-Rank Adaptation (LoRA) and introduce a novel magnitude-driven initialization strategy, LoRAM, that bridges the gap between efficiency and performance. Our work demystifies the prevailing awareness surrounding spectral initialization methods, demonstrating that their success primarily stems from the amplification of weight update magnitudes. By focusing on magnitude regulation as the key driver of convergence, we provide a unified perspective that connects seemingly disparate hyperparameter adjustments, such as learning rate, scaling factor, and initialization schemes, under a single framework.

**Limitations and Future Work.** Despite the advancements introduced in this work, several challenges remain open for future research. First, LoRAM mimics spectral initialization magnitudes rather than seeking optimal ones; exploring alternative strategies could yield further gains. Additionally, different layers may benefit from tailored magnitude settings, motivating joint optimization with learning rate and rank. Finally, our work does not explicitly address optimization dynamics and convergence properties. These directions remain valuable for advancing parameter-efficient fine-tuning.

---

[3]See Proposition 5 in Appendix. We prove LoRA-GA initialization maximizes LoRA gradient magnitude.

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

# A  Related Work

**Advances in Low-Rank Adaptation.**  The growing scale [1, 2, 3, 4, 64, 65] of large pre-trained models across various domains urgently demands efficient fine-tuning methods [18, 19, 20, 21, 22, 66, 67, 68, 69]. LoRA [7] has garnered significant attention in leveraging low-rank structures to represent weight updates, with subsequent studies expanding its foundations and applications [70, 48]. Existing improvements to LoRA primarily focus on three directions:

• The low-rank constraint creates bottlenecks when learning complex features. Recent approaches enhance expressiveness through iterative stacking of LoRA modules [27, 28, 24, 29], high-order matrix operations [23, 10], and customized architectural designs [71, 26, 72, 73, 74, 32, 75].
• The non-convex landscapes pose challenges for numerical optimization and hyperparameter tuning [30]. Researchers have explored to improve scaling factor [34, 76, 14], learning rate [35], dropout rate [77], optimizers [30, 38, 39, 44], and initialization strategies [40, 41, 42, 37, 39, 43, 44].
• Recent works reduce computational and memory overhead through freezing parameters [78, 36], designing more compact adapters [45, 79, 25] and parameter quantization [40, 46, 80].

**Knowledge-Driven Low-Rank Initialization.**  Proper initialization critically influences neural network training outcomes [81, 49]. The standard LoRA [7] initializes its low-rank matrices with random noise and zeros, referred to as "Noise & Zero" scheme, demonstrated to hinder convergence. Recent advances address this limitation by leveraging knowledge from pre-trained weights [40, 41, 42, 12] or task-specific data [39, 43, 44]. Most prominent approaches involve Singular Value Decomposition (SVD), which allows flexible control over matrix rank. As a pioneer, PiSSA [40] initializes LoRA weights using principal singular components of pre-trained matrices, aligning adaptation directions with the most significant parameter variations to accelerate convergence. Subsequent works [43, 44, 39, 82] propose data-driven strategies that incorporate domain knowledge into adapter construction. For instance, LoRA-GA [44] aligns low-rank gradient directions with full fine-tuning counterparts during initialization. While these methods boost performance and retain the core LoRA structure, their theoretical effects on optimization dynamics remain unclear. More importantly, they are less pluggable than LoRA, requiring extra computational pipelines and storage for SVD buffers.

**Optimization Dynamics of LoRA.** LoRA exhibits inherently nonlinear and non-convex optimization dynamics that complicate theoretical analysis [30]. Most existing theoretical studies are limited to simplified and idealized scenarios, such as the lazy-training regime [83, 84] and infinite-width limit[35, 37]. A common goal across existing studies involves ensuring feature learning stability to prevent unstable or collapsed training dynamics. For instance, [34] proves that improper scaling factors induce gradient collapse in high-rank adapters, proposing modified scaling mechanisms to stabilize forward and backward propagation. Recent work [35, 36, 37] reveals critical asymmetries in LoRA: The two low-rank matrices exhibit divergent distinct impacts on optimization trajectories. This asymmetry motivates [35] to employ distinct learning rates for each matrix. For SVD-based initialization, [39] investigates principal components derived from single-step full fine-tuning gradients [44], providing convergence guarantees yet under restrictive assumptions. Moreover, such analyses still fail to explain the efficacy of alternative approaches like weight-driven initialization [40].

The intricate nature of LoRA and its numerous advancements motivates us to seek a streamlined principle explaining its empirical success and guiding practical applications. We identify update magnitude amplification as a key mechanism, unifying seemingly disparate elements, such as scaling factors [34, 14], initialization strategies [40, 44], and learning rates [35], into a cohesive perspective.

# B  Lower Bound on Representation Error

**Proposition 1** (Lower Bound on Representation Error). *Consider the function class of LoRA-parameterized linear models:*

$$\mathcal{H} = \left\{ (W + \alpha BA)x \mid B \in \mathbb{R}^{n \times r}, A \in \mathbb{R}^{r \times m}, \nu[A] \leq M_1, \nu[B] \leq M_2 \right\}.$$

*Let $R(f) = \mathbb{E}_{(x,y) \sim \mathcal{D}}[\ell(f(x), y]$ denote the expected regression loss under data distribution $\mathcal{D}$, with squared error loss $\ell(f(x), y) = \|y - f(x)\|^2$. Define $f^* = \arg\min_f R(f)$ as the globally optimal predictor and $f_{\mathcal{H}}^* = \arg\min_{f \in \mathcal{H}} R(f)$ as the optimal predictor within $\mathcal{H}$. With loss of generality, we consider the optimal predictor is linear, i.e., $f^*(x) = W^*x$. Assume that the input covariance matrix is positive definite, i.e., $\Sigma_{\mathcal{D}} = \mathbb{E}_{x \sim \mathcal{D}}[xx^\top] \succ 0$, and the magnitudes is limited,*

satisfying $\alpha r\sqrt{mnM_1M_2} < \|W^* - W_0\|_F$. *Then, the representation error is strictly positive and lower bounded:*

$$R(f_{\mathcal{H}}^*) - R(f^*) \geq \lambda_{\min}(\Sigma_{\mathcal{D}}) \left( \|W^* - W_0\|_F - \alpha r\sqrt{mnM_1M_2} \right)^2 > 0,$$

*where $\lambda_{\min}(\cdot)$ denotes the smallest eigenvalue of the matrix.*

*Proof.* Due to the linear learnability, we have that $y(x) = f^*(x) = W^*x$. For any $W$, we have that

$$
\begin{aligned}
R(W) &= \mathbb{E}_{x\sim\mathcal{D}} \left[ \|(W - W^*)x\|^2 \right] \\
&= \mathbb{E}_{x\sim\mathcal{D}} \left[ (W - W^*)x \left( (W - W^*)x \right)^\top \right] \\
&= \text{Tr} \left( (W - W^*)\Sigma_{\mathcal{D}}(W - W^*)^\top \right) \\
&\geq \lambda_{\min}(\Sigma_{\mathcal{D}}) \|W - W^*\|_F^2.
\end{aligned}
\tag{13}
$$

The last inequality comes from $\lambda_{\min}(\Sigma_{\mathcal{D}}) > 0$.

Due to $\nu[A] \leq M_1, \nu[B] \leq M_2$, we have that

$$\|A\|_F^2 \leq rmM_1, \ \|B\|_F^2 \leq rnM_2.$$

Further, for any $A, \ B \in \mathcal{H}$, we have that

$$
\begin{aligned}
\|W_0 + \alpha BA - W^*\|_F &\geq \|W_0 - W^*\|_F - \|\alpha BA\|_F \\
&\geq \|W_0 - W^*\|_F - \alpha\|A\|_F\|B\|_F \\
&\geq \|W_0 - W^*\|_F - \alpha r\sqrt{mnM_1M_2} \\
&> 0.
\end{aligned}
\tag{14}
$$

Therefore, we have that

$$
\begin{aligned}
&R(f_{\mathcal{H}}^*) - R(f^*) \\
&= \min_{\nu[A]\leq M_1,\nu[B]\leq M_2} R(W_0 + \alpha BA) \\
&\geq \lambda_{\min}(\Sigma_{\mathcal{D}}) \min_{\nu[A]\leq M_1,\nu[B]\leq M_2} \|W_0 + \alpha BA - W^*\|_F^2 \\
&\geq \lambda_{\min}(\Sigma_{\mathcal{D}}) \left( \|W_0 - W^*\|_F - \alpha r\sqrt{mnM_1M_2} \right)^2 > 0.
\end{aligned}
$$

The first inequality comes from Eq. (13) and the last inequality comes from Eq. (14). □

## C  Proof of Main Theorems

### C.1  Proof of Parameter Scaling Equivalence

**Proposition 2** (Parameter Scaling Equivalence)**.** *For LoRA layers defined in Eq. (1), consider decomposing the scaling factor $\alpha = \alpha'\alpha_A\alpha_B$, where $\alpha', \alpha_A, \alpha_B \in \mathbb{R}^+$. Under the commonly used optimization frameworks with negligible numerical errors, the following parametrization schemes exhibit dynamical equivalence throughout training: For all iterations $t \geq 0$, $\Delta W_{\text{LoRA}}^{(t)} = \Delta \tilde{W}_{\text{LoRA}}^{(t)}$ and $W_{\text{LoRA}}^{(t)} = \tilde{W}_{\text{LoRA}}^{(t)}$, where $\tilde{A}^{(t)}$, $\tilde{B}^{(t)}$ and $\tilde{W}^{(t)}$ represent the re-parameterized versions.*

| | *Original* | *SGD* | *Adam* |
|---|---|---|---|
| *Representation* | $\alpha BAx$ | $\alpha'\tilde{B}\tilde{A}x$ | $\alpha'\tilde{B}\tilde{A}x$ |
| *Initialization* | $A^{(0)} = A_{init}, B^{(0)} = B_{init}$ | $\tilde{A}^{(0)} = \alpha_A A_{init}, \tilde{B}^{(0)} = \alpha_B B_{init}$ | $\tilde{A}^{(0)} = \alpha_A A_{init}, \tilde{B}^{(0)} = \alpha_B B_{init}$ |
| *Learning Rates* | $\eta_A > 0, \eta_B > 0$ | $\eta_{\tilde{A}} = \alpha_A^2\eta_A, \eta_{\tilde{B}} = \alpha_B^2\eta_B$ | $\eta_{\tilde{A}} = \alpha_A\eta_A, \eta_{\tilde{B}} = \alpha_B\eta_B$ |

*Proof.* Given LoRA's weight decomposition $W_{\text{LoRA}} = \alpha BA$ where $\alpha > 0$, the gradient updates follow:

$$\Delta W_{\text{LoRA}}^{(t)} = \alpha(B^{(t+1)}A^{(t+1)} - B^{(t)}A^{(t)}), \tag{15}$$

with learning rates $\eta_A, \eta_B$ for parameters $A$ and $B$, respectively. We define reparameterized parameters $\tilde{A} = \alpha_A A$, $\tilde{B} = \alpha_B B$, and $\alpha' = \alpha/(\alpha_A\alpha_B)$. The LoRA projection becomes:

$$W_{\text{LoRA}} = \alpha BA = \alpha'\tilde{B}\tilde{A}. \tag{16}$$

This transformation preserves the functional form while redistributing scaling factors.

**Proof for SGD**  Under initial conditions:

$$\tilde{A}^{(0)} = \alpha_A A_{\text{init}}, \tag{17}$$

$$\tilde{B}^{(0)} = \alpha_B B_{\text{init}}. \tag{18}$$

Gradients for reparameterized parameters:

$$\nabla_{\tilde{A}} L = \alpha' \tilde{B}^\top \nabla_W L, \tag{19}$$

$$\nabla_{\tilde{B}} L = \alpha' \nabla_W L \tilde{A}^\top. \tag{20}$$

With learning rates $\eta_{\tilde{A}} = \alpha_A^2 \eta_A$, $\eta_{\tilde{B}} = \alpha_B^2 \eta_B$ and $t = 0$:

$$\tilde{A}^{(t+1)} = \tilde{A}^{(t)} - \alpha_A^2 \eta_A \nabla_{\tilde{A}} L^{(t)} \tag{21}$$

$$= \alpha_A A^{(t)} - \alpha_A^2 \eta_A (\alpha' \alpha_B B^{(t)})^\top \nabla_W L^{(t)} \tag{22}$$

$$= \alpha_A \left( A^{(t)} - \alpha \eta_A (B^{(t)})^\top \nabla_W L^{(t)} \right) \tag{23}$$

$$= \alpha_A \left( A^{(t)} - \eta_A \nabla_A L^{(t)} \right) \tag{24}$$

$$= \alpha_A A^{(t+1)}. \tag{25}$$

Similarly for $\tilde{B}$:

$$\tilde{B}^{(t+1)} = \alpha_B \left( B^{(t)} - \eta_B \nabla_B L^{(t)} \right) = \alpha_B B^{(t+1)}. \tag{26}$$

These equations propagate through all $t > 0$ via mathematical induction. Compare weight increments:

$$\Delta \tilde{W}_{\text{LoRA}}^{(t)} = \alpha' [\tilde{B}^{(t+1)} \tilde{A}^{(t+1)} - \tilde{B}^{(t)} \tilde{A}^{(t)}] \tag{27}$$

$$= \alpha' \alpha_A \alpha_B [B^{(t+1)} A^{(t+1)} - B^{(t)} A^{(t)}] \tag{28}$$

$$= \alpha [B^{(t+1)} A^{(t+1)} - B^{(t)} A^{(t)}] \tag{29}$$

$$= \Delta W_{\text{LoRA}}^{(t)}. \tag{30}$$

**Proof for Adam**  Adam maintains exponential moving averages $(m_t, v_t)$ for gradients. Under parameter scaling $\tilde{P} = kP$, gradients transform as $\nabla_{\tilde{P}} L = k \nabla_P L$. The momentum terms inherit scaling factors:

$$m_{\tilde{P}} = \beta_1 m_{\tilde{P}} + (1 - \beta_1) k \nabla_P L, \tag{31}$$

$$v_{\tilde{P}} = \beta_2 v_{\tilde{P}} + (1 - \beta_2) k^2 (\nabla_P L)^2. \tag{32}$$

The pratical gradients for Adam:

$$\nabla^\dagger \tilde{P} = m_{\tilde{P}} / \sqrt{v_{\tilde{P}} + \epsilon} \tag{33}$$

$$= [\beta_1 m_P + ...] / \sqrt{\beta_2 v_P + ...} \tag{34}$$

When setting $\epsilon = 0$ for alleviating numerical errors, we have $\nabla^\dagger \tilde{P} = \nabla^\dagger P$. Setting $\eta_{\tilde{P}} = k \eta_P$ cancels scaling factors, preserving update magnitudes. With learning rates $\eta_{\tilde{A}} = \alpha_A \eta_A$, $\eta_{\tilde{B}} = \alpha_B \eta_B$ and $t = 0$:

$$\tilde{A}^{(t+1)} = \tilde{A}^{(t)} - \alpha_A \eta_A \nabla_{\tilde{A}}^\dagger L^{(t)} \tag{35}$$

$$= \alpha_A A^{(t)} - \alpha_A \eta_A \nabla_A^\dagger L^{(t)} \tag{36}$$

$$= \alpha_A A^{(t+1)}. \tag{37}$$

Similarly for $\tilde{B}$. These equalities propagate through all $t > 0$ via mathematical induction.

$$\square$$

## C.2   Proof of Proposition 2

For the convenience, we split the proof of Proposition 2 into two parts.

**Proposition 3** (Parameter Magnitude Dynamics)**.** *Consider LoRA parameters $A^{(t)}$ and $B^{(t)}$ updated with learning rate $\eta$. Assume: $A^{(0)} \sim \mathcal{N}(\mathbf{0}, \sigma_A^2 I)$, $B^{(0)} \sim \mathcal{N}(\mathbf{0}, \sigma_B^2 I)$, $\nabla_W L \sim \mathcal{N}(\mathbf{0}, \sigma_L^2 I)$, and $\mathbb{E}[\langle A^{(t)}, \nabla_A L^{(t)} \rangle] = \mathbb{E}[\langle B^{(t)}, \nabla_B L^{(t)} \rangle] = 0$. Under these conditions, the parameter norm vector $\mathbf{e}_t = \left[ \mathbb{E}[\|A^{(t)}\|_F^2], \mathbb{E}[\|B^{(t)}\|_F^2] \right]^T$ evolve as the following linear dynamical system (left) with closed-form solution (right):*

$$\mathbf{e}_{t+1} = \left( I + \begin{bmatrix} 0 & \gamma_A \\ \gamma_B & 0 \end{bmatrix} \right) \mathbf{e}_t, \quad \mathbf{e}_t = \begin{bmatrix} \frac{\lambda_+^t + \lambda_-^t}{2} & \sqrt{\frac{\gamma_A}{\gamma_B}} \frac{\lambda_+^t - \lambda_-^t}{2} \\ \sqrt{\frac{\gamma_B}{\gamma_A}} \frac{\lambda_+^t - \lambda_-^t}{2} & \frac{\lambda_+^t + \lambda_-^t}{2} \end{bmatrix} \mathbf{e}_0. \tag{38}$$

*where $\gamma_A = m\eta^2 \sigma_L^2$, $\gamma_B = n\eta^2 \sigma_L^2$ and $\lambda_\pm = 1 \pm \sqrt{\gamma_A \gamma_B}$. Similarly, the parameter magnitudes $\boldsymbol{\nu}_t = \left[ \mathbb{E}[\nu[A^{(t)}]], \mathbb{E}[\nu[B^{(t)}]] \right]^T$ evolve as a linear dynamical system:*

$$\boldsymbol{\nu}_t = \left( I + \begin{bmatrix} 0 & \gamma_B \\ \gamma_A & 0 \end{bmatrix} \right) \boldsymbol{\nu}_{t-1}. \tag{39}$$

*Proof.* Following the gradient descent update rule, the parameter dynamics are:

$$\begin{cases} A_{t+1} = A_t - \eta \nabla_A L_t \\ B_{t+1} = B_t - \eta \nabla_B L_t \end{cases}, \tag{40}$$

where the gradient relationships $\nabla_A L = B^\top (\nabla_W L)$ and $\nabla_B L = (\nabla_W L) A^\top$ are derived from the LoRA architecture. Expand the Frobenius norm for $A_{t+1}$:

$$\|A_{t+1}\|_F^2 = \|A_t\|_F^2 - 2\eta \langle A_t, \nabla_A L_t \rangle + \eta^2 \|\nabla_A L_t\|_F^2. \tag{41}$$

Taking expectations (using independence $\mathbb{E}[\langle A_t, \nabla_A L_t \rangle] = 0$):

$$\mathbb{E}\|A_{t+1}\|_F^2 = \mathbb{E}\|A_t\|_F^2 + \eta^2 \mathbb{E}\|\nabla_A L_t\|_F^2. \tag{42}$$

Similarly, for $B_{t+1}$:

$$\mathbb{E}\|B_{t+1}\|_F^2 = \mathbb{E}\|B_t\|_F^2 + \eta^2 \mathbb{E}\|\nabla_B L_t\|_F^2. \tag{43}$$

Expand the norm of $\nabla_A L = B^\top (\nabla_W L)$:

$$\|\nabla_A L\|_F^2 = \text{Tr}((\nabla_A L)(\nabla_A L)^\top) = \text{Tr}(B^\top (\nabla_W L)(\nabla_W L)^\top B). \tag{44}$$

Taking expectations (using gradient entry independence):

$$\mathbb{E}\|\nabla_A L\|_F^2 = \text{Tr}\left( B^\top \mathbb{E}[(\nabla_W L)(\nabla_W L)^\top] B \right) = m\sigma_L^2 \|B\|_F^2. \tag{45}$$

where $\mathbb{E}[(\nabla_W L)(\nabla_W L)^\top] = m\sigma_L^2 I_n$ follows from the i.i.d. assumption on gradient entries.

For $\nabla_B L = (\nabla_W L) A^\top$:

$$\mathbb{E}\|\nabla_B L\|_F^2 = \text{Tr}\left( A\mathbb{E}[(\nabla_W L)^\top (\nabla_W L)] A^\top \right) = n\sigma_L^2 \|A\|_F^2. \tag{46}$$

Substitute gradient norms results:

$$\begin{cases} \mathbb{E}\|A_{t+1}\|_F^2 = \mathbb{E}\|A_t\|_F^2 + \eta^2 m\sigma_L^2 \mathbb{E}\|B_t\|_F^2 \\ \mathbb{E}\|B_{t+1}\|_F^2 = \mathbb{E}\|B_t\|_F^2 + \eta^2 n\sigma_L^2 \mathbb{E}\|A_t\|_F^2 \end{cases}. \tag{47}$$

Define parameters and state vector:

$$\gamma_A = \eta^2 m\sigma_L^2, \quad \gamma_B = \eta^2 n\sigma_L^2, \quad \mathbf{e}_t = \begin{bmatrix} \mathbb{E}\|A_t\|_F^2 \\ \mathbb{E}\|B_t\|_F^2 \end{bmatrix}. \tag{48}$$

The recursive system becomes:

$$\mathbf{e}_{t+1} = \begin{bmatrix} 1 & \gamma_A \\ \gamma_B & 1 \end{bmatrix} \mathbf{e}_t = \left( I + \begin{bmatrix} 0 & \gamma_A \\ \gamma_B & 0 \end{bmatrix} \right) \mathbf{e}_t. \tag{49}$$

Similar results for $\boldsymbol{\nu}_t$ can be obtained using the same proof techniques.

$\square$

**Proposition 4** (Linearized Dynamics Approximation). *For sufficiently small learning rate $\eta$, the closed-form solution admits the first-order approximation:*

$$\boldsymbol{\nu}_t \approx \left( I + t \begin{bmatrix} 0 & \gamma_B \\ \gamma_A & 0 \end{bmatrix} \right) \boldsymbol{\nu}_0, \quad \text{where } \boldsymbol{\nu}_t \triangleq \begin{bmatrix} \nu[A^{(t)}] \\ \nu[B^{(t)}] \end{bmatrix} = \begin{bmatrix} \sigma_A^2 + t\gamma_B \sigma_B^2 \\ \sigma_B^2 + t\gamma_A \sigma_A^2 \end{bmatrix}. \tag{50}$$

*This yields the update variance expansion under small-$\eta$ regime:*

$$\nu[W_{LoRA}^{(t)}] \approx k_1 \gamma t + k_2 \gamma^2 t^2, \tag{51}$$

*with $\gamma = \eta^2 \sigma_L^2$, $k_1 = r(m\nu[A^{(0)}]^2 + n\nu[B^{(0)}]^2)$, and $k_2 = rmn\nu[A^{(0)}]\nu[B^{(0)}]$.*

*Proof.* For a sufficiently small learning rate $\eta$, the product $\gamma_A \gamma_B = mn\eta^4 \sigma_L^4$ is very small. Hence, the eigenvalues

$$\lambda_\pm = 1 \pm \sqrt{\gamma_A \gamma_B} \tag{52}$$

can be approximated via a first-order Taylor expansion. Thus, for small $t$, the closed-form solution for $\boldsymbol{\nu}_t$ can be approximated as

$$\boldsymbol{\nu}_t \approx \left( I + t \begin{bmatrix} 0 & \gamma_B \\ \gamma_A & 0 \end{bmatrix} \right) \boldsymbol{\nu}_0. \tag{53}$$

By definition, we set

$$\boldsymbol{\nu}_t \triangleq \begin{bmatrix} \nu[A^{(t)}] \\ \nu[B^{(t)}] \end{bmatrix} = \begin{bmatrix} \sigma_A^2 + t\,\gamma_B \sigma_B^2 \\ \sigma_B^2 + t\,\gamma_A \sigma_A^2 \end{bmatrix}, \tag{54}$$

which agrees with the first-order expansion of the linear system. $\qquad\square$

# D   Analyzing LoRA-GA from Magnitude Principle

**Proposition 5** (LoRA-GA initialization maximizes low-rank gradient magnitude). *Given a gradient matrix $\nabla_W L \in \mathbb{R}^{n \times m}$ with SVD decomposition $\nabla_W L = USV^\top$, the optimal rank-$r$ matrices $A \in \mathbb{R}^{r \times m}$ and $B \in \mathbb{R}^{n \times r}$ that maximize the Frobenius norm of the gradients $\|\nabla_A L\|_F^2$ and $\|\nabla_B L\|_F^2$ in Eq. (2), under the constraints $\|A\|_F^2 = r$ and $\|B\|_F^2 = r$, are given by:*

$$A^\star = V_{:,:r}^\top, \quad B^\star = U_{:,:r}, \tag{55}$$

*where $V_{:,:r}$ and $U_{:,:r}$ contain the first $r$ right and left singular vectors of $\nabla_W L$, respectively.*

*Proof.* Let $G = \nabla_W L \in \mathbb{R}^{n \times m}$ and write its SVD $G = USV^\top$ with singular values $\sigma_1 \geq \sigma_2 \geq \cdots \geq 0$. From the LoRA gradient expressions we have

$$\nabla_A L = B^\top G, \qquad \nabla_B L = GA^\top.$$

Note that $\nabla_A L$ does not depend on $A$ and $\nabla_B L$ does not depend on $B$, hence the two maximization problems decouple.

*(i) Maximizing $\|\nabla_B L\|_F^2$ w.r.t. $A$ under $\|A\|_F^2 = r$.* Let $A' := AV$. Then $\|A'\|_F^2 = \|A\|_F^2 = r$ and

$$\|GA^\top\|_F^2 = \|US(A')^\top\|_F^2 = \|S(A')^\top\|_F^2 = \mathrm{tr}\big(A'S^2(A')^\top\big).$$

Form the Lagrangian $\mathcal{L}(A', \lambda) = \mathrm{tr}(A'S^2(A')^\top) - \lambda(\mathrm{tr}(A'(A')^\top) - r)$. Taking derivative w.r.t. $A'$ yields the stationarity condition $A'S^2 = \lambda A'$. Thus the rows of $A'$ lie in the eigenspaces of $S^2$. To maximize the trace, one places the Frobenius norm onto the coordinates corresponding to the largest diagonal entries of $S^2$, i.e. the first $r$ singular values. Taking $A'^\star = [I_r \mid 0]$ attains the maximum

$$\max_{\|A\|_F^2 = r} \|GA^\top\|_F^2 = \sum_{i=1}^r \sigma_i^2,$$

and $A^\star = A'^\star V^\top = V_{:,:r}^\top$ (up to orthonormal transformations within the chosen $r$-subspace).

*(ii) Maximizing $\|\nabla_A L\|_F^2$ w.r.t. $B$ under $\|B\|_F^2 = r$.* Analogously let $B' := U^\top B$. Then

$$\|B^\top G\|_F^2 = \mathrm{tr}\big((B')^\top S^2 B'\big),$$

and the same Lagrange argument gives that the optimal choice concentrates norm on the first $r$ coordinates, producing

$$\max_{\|B\|_F^2 = r} \|B^\top G\|_F^2 = \sum_{i=1}^{r} \sigma_i^2,$$

and one may take $B^\star = U_{:,:r}$.

Hence the stated solutions $A^\star = V_{:,:r}^\top$ and $B^\star = U_{:,:r}$ are optimal, with the above maximal values. This completes the proof. $\qquad\square$

## E  Details on Key Formulas

### E.1  Weight Update Magnitude

$$\nu[\Delta W_{\text{LoRA}}^{(t)}] \approx r\alpha^2\eta^2 \left( \nu[B^{(t)}]\nu[\nabla_A L^{(t)}] + \nu[\nabla_B L^{(t)}]\nu[A^{(t)}] \right).$$

*Proof.* Given the LoRA parameter update rule:

$$\Delta W_{\text{LoRA}}^{(t)} = \alpha\eta \left( B^{(t)}\nabla_A L^{(t)} + \nabla_B L^{(t)} A^{(t)} \right) + \mathcal{O}(\eta^2), \tag{56}$$

where we retain first-order terms in $\eta$ under small learning rate assumption.

**Assumptions:**

(A1) Independence: $B \perp\!\!\!\perp \nabla_A L, \nabla_B L \perp\!\!\!\perp A$

(A2) Zero-mean initialization: $\mathbb{E}[B_{ij}] = \mathbb{E}[A_{kl}] = 0$

(A3) Spatial homogeneity: $var[B_{ij}] = \nu[B]$, $var[(\nabla_A L)_{ij}] = \nu[\nabla_A L]$.

For entry $(m, n)$ in $\Delta W_{\text{LoRA}}^{(t)}$:

$$var\left[(\Delta W)_{mn}\right] \approx \alpha^2\eta^2 var\left[\sum_{k=1}^{r} \left(B_{mk}(\nabla_A L)_{kn} + (\nabla_B L)_{mk}A_{kn}\right)\right] \tag{57}$$

$$= \alpha^2\eta^2 \left( var\left[\sum_k B_{mk}(\nabla_A L)_{kn}\right] + var\left[\sum_k (\nabla_B L)_{mk}A_{kn}\right] \right). \tag{58}$$

By assumptions (A1)-(A3):

$$var\left[\sum_k B_{mk}(\nabla_A L)_{kn}\right] = \sum_k var[B_{mk}]var[(\nabla_A L)_{kn}] \tag{59}$$

$$= r\nu[B]\nu[\nabla_A L]. \tag{60}$$

Similarly:

$$var\left[\sum_k (\nabla_B L)_{mk}A_{kn}\right] = r\nu[\nabla_B L]\nu[A]. \tag{61}$$

Combining these terms:

$$\nu[\Delta W_{\text{LoRA}}^{(t)}] \approx \alpha^2\eta^2 \left( r\nu[B]\nu[\nabla_A L] + r\nu[\nabla_B L]\nu[A] \right) \tag{62}$$

$$= r\alpha^2\eta^2 \left( \nu[B]\nu[\nabla_A L] + \nu[\nabla_B L]\nu[A] \right). \tag{63}$$

where $\nu[B]$, $\nu[A]$ inherit their magnitudes from initialization scheme, and $\nu[\nabla L]$ terms reflect task-specific loss landscape characteristics. $\qquad\square$

## E.2 Spectral Concentration Factor

Let the singular values of the pretrained weight matrix $W$ be $\{s_i\}_{i=1}^{\mathcal{R}[W]}$. Define the average of the top-$r$ singular values as

$$\mathbb{E}_r[s] = \frac{1}{r} \sum_{i=1}^{r} s_i, \tag{64}$$

and the average of the squares of all singular values as

$$\mathbb{E}_{\mathcal{R}[W]}[s^2] = \frac{1}{\mathcal{R}[W]} \sum_{i=1}^{\mathcal{R}[W]} s_i^2. \tag{65}$$

Because the square function is convex, Jensen's inequality implies

$$\mathbb{E}_r[s]^2 \leq \mathbb{E}_r[s^2], \tag{66}$$

with equality if and only if all $s_i$ (for $i = 1, \ldots, r$) are equal. Moreover, since lower singular values generally contribute less to the overall energy, as $r$ increases the value of $\mathbb{E}_r[s]$ decreases relative to $\mathbb{E}_{\mathcal{R}[W]}[s^2]$, making $\rho[r]$ a monotonically decreasing function of $r$.

## E.3 Variance of $A_{\text{SVD}}$ and $B_{\text{SVD}}$

Recall that the PiSSA initialization is given by

$$A_{\text{SVD}} = \sqrt{S_r}\, V_{:,:r}^{\top}, \quad B_{\text{SVD}} = U_{:,:r}\sqrt{S_r}, \tag{67}$$

where $S_r$ is a diagonal matrix containing the top-$r$ singular values of $W$. Assuming that the columns of $V$ (and similarly, $U$) form an orthonormal basis, the variance of $V_{:,:r}$ (taken elementwise) is approximately $\nu[V_{:r}] \approx \frac{1}{m}$ (or $\frac{1}{n}$ for $U$), since for an orthogonal matrix the energy is uniformly distributed. Thus, the variance of $A_{\text{SVD}}$ can be expressed as:

$$\nu(A_{\text{SVD}}) = \frac{\text{Tr}(\sqrt{S_r^T}V_{:,:r}^T V_{:,:r}\sqrt{S})}{mr} = \frac{\sum_{i=1}^{r} s_i}{mr} = \mathbb{E}_r[s]\nu[V_{:r}]. \tag{68}$$

To connect this with the variance of $W$, note that

$$\nu[W] = \frac{1}{mn} \sum_{i=1}^{\mathcal{R}[W]} s_i^2. \tag{69}$$

Thus, we can relate $\mathbb{E}_r[s]$ to $\nu[W]$ via the spectral concentration factor $\rho[r]$. Incorporating a factor of $n$ to account for the dimensions of $A_{\text{SVD}}$, we obtain:

$$\nu(A_{\text{SVD}}) = \sqrt{\frac{n\,\rho[r]\,\nu[W]}{m\,\mathcal{R}[W]}}. \tag{70}$$

A similar argument, with the roles of $m$ and $n$ interchanged, leads to the expression for $\nu(B_{\text{SVD}})$.

## E.4 Spectral Gain Factor

The dynamics of the LoRA weight update variance are captured by an expression of the form:

$$\nu[W_{\text{LoRA}}^{(t)}] \approx k_1\gamma t + k_2\gamma^2 t^2, \tag{71}$$

where $k_1$ is the linear evolution rate. Substituting the variance expressions derived for $A_{\text{SVD}}$ and $B_{\text{SVD}}$ into the dynamical system (see Eq. (6)), we obtain:

$$k_1 = \frac{\rho[r]\,r\,(m+n)}{\mathcal{R}[W]}\,\nu[W]. \tag{72}$$

Defining

$$Q[r] \triangleq \frac{\rho[r]\,r}{\mathcal{R}[W]}, \tag{73}$$

this expression becomes:

$$k_1 = Q[r](m+n)\nu[W], \tag{74}$$

with the constraint $0 \leq Q[r] \leq 1$. When $\nu[W] \sim \mathcal{O}(\min(1/m, 1/n))$, the factor $Q[r]$ effectively quantifies the amplification of the weight update magnitude due to the spectral initialization.

*Proof.* Recall that

$$\rho[r] \triangleq \frac{\left(\frac{1}{r}\sum_{i=1}^{r} s_i\right)^2}{\frac{1}{\mathcal{R}[W]}\sum_{i=1}^{\mathcal{R}[W]} s_i^2},$$ (75)

and the spectral gain factor is defined as

$$Q[r] \triangleq \frac{\rho[r]\, r}{\mathcal{R}[W]}.$$ (76)

Let $m = \mathcal{R}[W]$. By Jensen's inequality (or by the Cauchy–Schwarz inequality), we have

$$\left(\frac{1}{r}\sum_{i=1}^{r} s_i\right)^2 \leq \frac{1}{r}\sum_{i=1}^{r} s_i^2.$$ (77)

Therefore,

$$\rho[r] \leq \frac{\frac{1}{r}\sum_{i=1}^{r} s_i^2}{\frac{1}{m}\sum_{i=1}^{m} s_i^2} = \frac{m}{r}\cdot\frac{\sum_{i=1}^{r} s_i^2}{\sum_{i=1}^{m} s_i^2} \leq \frac{m}{r},$$ (78)

since $\frac{\sum_{i=1}^{r} s_i^2}{\sum_{i=1}^{m} s_i^2} \leq 1$. Substituting this bound into the definition of $Q[r]$ yields

$$Q[r] = \frac{\rho[r]\, r}{m} \leq \frac{m}{r}\cdot\frac{r}{m} = 1.$$ (79)

This completes the proof that $Q[r] \leq 1$. □

# F  Further Clarifications

In this section, we provide additional details regarding the experimental setup for our theoretical validations and justify the core assumptions underlying our propositions.

## F.1  Clarification on Figure 2: Experimental Setup and Optimizer Dynamics

The experiments illustrated in Figure 2 were conducted in a controlled environment to isolate and validate our theoretical claims regarding hyperparameter equivalence and magnitude dynamics. The setup uses a 5-layer MLP with an intermediate dimension of 400, where LoRA modules are trained to fit a mapping for randomly generated synthetic data. This simple setting effectively tests the fundamental fitting capabilities of LoRA.

Figure 2(a) is specifically designed to empirically validate Proposition 1, reproduced with clearer separation in Figure 6. We deliberately use two different optimizers, SGD for the first 2,500 steps and Adam thereafter, to demonstrate that when the hyperparameter product $\alpha'\alpha_A\alpha_B$ is held constant, LoRA exhibits equivalent training trajectories regardless of the optimizer. This equivalence is confirmed by the overlapping loss curves. Furthermore, the norm difference $||\Delta\tilde{W}_{\text{LoRA}}^{(t)} - \Delta W_{\text{LoRA}}^{(t)}||_F^2$ between the baseline and other equivalent settings remains near zero, confirming that the learned weights themselves evolve identically.

## F.2  Clarification on Figure 3: Direct Relationship to PiSSA

The goal of Figure 3 is to visualize the source of magnitude gain in spectral initialization methods, for which we use PiSSA [40] as a representative example. The link is direct:

• The solid colored curves for the spectral concentration factor ($\rho[r]$) and spectral gain factor ($Q[r]$) are calculated directly from the SVD of the pretrained weights. This is precisely the mechanism that PiSSA employs for initialization. Therefore, these curves illustrate the inherent magnitude dynamics of a PiSSA-initialized model.

• The plots reveal two key insights from our analysis of PiSSA. First, the plot of $\rho[r]$ shows that spectral energy is highly concentrated in the top singular values, explaining why PiSSA is particularly effective at low ranks. Second, the plot of $Q[r]$ quantifies the significant magnitude gain that PiSSA (colored lines) achieves over the naive "Noise & Zeros" baseline (white dotted line). This is precisely the gain that LoRAM is designed to mimic efficiently without performing SVD.

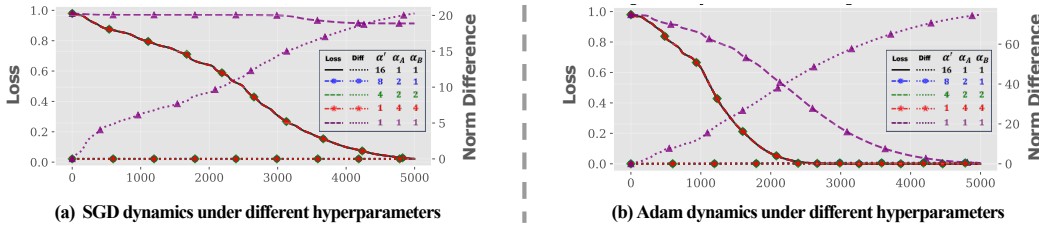

(a) SGD dynamics under different hyperparameters      (b) Adam dynamics under different hyperparameters

Figure 6: A detailed view of the validation of Proposition 1, separating the SGD and Adam optimization phases. Each curve represents a model with a unique hyperparameter combination. The norm difference (right axis) aggregates the Frobenius norm discrepancies between the baseline model (black curve) and others across all layers. The results show that diverse hyperparameter sets can produce identical optimization trajectories, confirming our theoretical equivalence framework.

# G    Further Analysis on the Interplay of Initialization, Learning Rate, and Performance

Our primary experiments were conducted under a unified hyperparameter configuration to ensure a fair and controlled comparison. However, different initialization strategies may achieve optimal performance under varying hyperparameters, particularly the learning rate. To provide a more nuanced understanding, we conducted a supplementary analysis investigating the performance of LoRAM, PiSSA [40], and MiLoRA [41] under different learning rates.

Specifically, we compare the results from our original experiments using a moderate learning rate ($2 \times 10^{-5}$) with new results obtained using a higher learning rate ($2 \times 10^{-4}$), which aligns with the setting used in the original MiLoRA paper. The detailed results for ranks $r = 16$ and $r = 128$ are presented in Table 5 and Table 6, respectively.

Table 5: Performance of PiSSA, LoRAM, and MiLoRA with different learning rates at rank $r = 16$.

| Learning rate | $2 \times 10^{-5}$ | | | $2 \times 10^{-4}$ | | |
|---|---|---|---|---|---|---|
| **Method** | PiSSA | LoRAM | MiLoRA | PiSSA | LoRAM | MiLoRA |
| GSM8K | 37.68 | 40.32 | 29.70 | 52.46 | 53.29 | 46.62 |
| MATH | 5.16 | 5.30 | 4.18 | 8.80 | 8.92 | 6.18 |
| HumanEval | 18.37 | 18.92 | 14.69 | 24.47 | 25.62 | 17.14 |
| MBPP | 28.62 | 28.83 | 27.23 | 31.22 | 31.74 | 28.65 |
| Commonsense | 73.72 | 75.19 | 67.90 | 77.02 | 79.11 | 76.67 |

Table 6: Performance of PiSSA, LoRAM, and MiLoRA with different learning rates at rank $r = 128$.

| Learning rate | $2 \times 10^{-5}$ | | | $2 \times 10^{-4}$ | | |
|---|---|---|---|---|---|---|
| **Method** | PiSSA | LoRAM | MiLoRA | PiSSA | LoRAM | MiLoRA |
| GSM8K | 51.48 | 51.12 | 39.81 | 58.37 | 59.28 | 54.66 |
| MATH | 7.04 | 7.25 | 5.18 | 11.46 | 10.76 | 9.20 |
| HumanEval | 21.62 | 22.03 | 20.39 | 26.81 | 29.95 | 28.02 |
| MBPP | 31.07 | 31.53 | 29.95 | 36.54 | 37.80 | 33.35 |
| Commonsense | 77.28 | 77.81 | 74.29 | 67.09 | 74.23 | 79.01 |

The results reveal several key patterns:

• **At a lower rank** ($r = 16$), all methods generally benefit from the higher learning rate, showing improved performance across tasks. This suggests that in low-rank settings, a larger learning rate can enhance convergence speed.

• **At a higher rank** ($r = 128$), a clear divergence emerges. While most methods still improve, PiSSA exhibits a notable performance degradation on the Commonsense dataset. Its training loss

curve in this high-rank, high-LR configuration plateaus early, suggesting that the amplified updates overshoot the effective descent direction.

• These findings align perfectly with our **magnitude principle**. PiSSA's use of principal singular components leads to stronger initial magnitude amplification, which necessitates a smaller optimal learning rate to maintain stability. Conversely, MiLoRA, which initializes with smaller minor singular components, can benefit from a larger learning rate. This analysis reinforces our claim from Section 4.3: "Magnitude scaling should be applied conservatively at higher ranks, since larger ranks inherently amplify updates."

This supplementary analysis underscores that the optimal learning rate is intrinsically linked to the magnitude scaling introduced by the initialization method.

## H   Efficiency Analysis: Computational and Memory Overhead

A primary motivation for LoRAM is to achieve the performance gains of spectral initialization methods while preserving the computational efficiency and minimal memory footprint of the original LoRA framework. In this section, we provide a detailed comparison of the theoretical and practical overhead associated with LoRAM, LoRA, and PiSSA.

### H.1   Theoretical Complexity

The theoretical time and space complexities for the initialization phase of each method are summarized in Table 7. LoRA's overhead is minimal, stemming from random matrix initialization. LoRAM maintains this same linear complexity, as its deterministic DST basis is generated efficiently through vector multiplication. In contrast, PiSSA's complexity is dominated by the SVD of the pretrained weight matrices, a computationally intensive operation. Furthermore, PiSSA requires storing both the original and decomposed components, effectively doubling the space requirement compared to LoRA and LoRAM.

Table 7: Theoretical initialization time and space complexities of LoRA, LoRAM, and PiSSA.

| Method | LoRA | LoRAM | PiSSA |
|---|---|---|---|
| Time | $O(mr + nr)$ | $O(mr + nr)$ | $O(\min(m^2 n, mn^2))$ |
| Space | $O(r(m + n))$ | $O(r(m + n))$ | $O(2r(m + n))$ |

### H.2   Practical Performance

We empirically validated these complexities by measuring the practical initialization time and memory usage while fine-tuning LLaMA-7B on an 8-GPU server. The results, shown in Table 8, highlight two distinct workflows for PiSSA:

1. **Pre-processing :** This approach first computes and saves the residual model after subtracting the low-rank approximation. While the initialization itself is faster, this process incurs a substantial one-time storage cost, requiring over 12.5 GB to store the residual weights in addition to the LoRA adapter weights.
2. **Direct Workflow:** This method computes SVD on-the-fly, which avoids the large storage overhead. However, it is significantly slower (often taking over 10 minutes in our setup) due to known bottlenecks related to CPU-based SVD computations before GPU transfer .

As shown in the table, LoRAM's practical performance is nearly identical to that of the standard LoRA implementation, demonstrating its exceptional efficiency . It successfully eliminates the significant time and space overhead introduced by SVD-based methods like PiSSA, confirming its utility as a lightweight yet powerful initialization strategy.

## I   Scope and Limitations of the Magnitude Principle

While our work establishes the magnitude of weight updates as a fundamental driver of LoRA's performance, it is crucial to clearly define the scope and limitations of this principle. Our central

Table 8: Practical initialization time and space cost of LoRA, LoRAM, and PiSSA on LLaMA-7B.

| Metric | LoRA | LoRAM | PiSSA (pre-process) | PiSSA (direct) |
|--------|------|-------|---------------------|----------------|
| Time ($r = 16$) | 1.36s | 0.95s | 51.32s | $> 10$min |
| Time ($r = 128$) | 4.23s | 2.19s | 57.79s | $> 10$min |
| Time ($r = 512$) | 8.50s | 5.19s | 110.31s | $> 10$min |
| Space ($r = 16$) | 152MB | 152MB | 12.5GB + 152MB | 305MB |
| Space ($r = 128$) | 1.2GB | 1.2GB | 12.5GB + 1.2GB | 2.4GB |
| Space ($r = 512$) | 4.8GB | 4.8GB | 12.5GB + 4.8GB | 9.6GB |

claim is that magnitude plays a **primary, not universal**, role in the success of various hyperparameter-tuning strategies. The principle provides a coherent and predictive lens for unifying these strategies, rather than suggesting that magnitude is the sole determinant of all outcomes.

The applicability and effects of magnitude scaling are subject to several important considerations:

• **Interplay with Other Hyperparameters:** As indicated by our analysis in Proposition 2, the magnitude principle is not limited to initialization magnitudes ($\sigma_A^2, \sigma_B^2$) alone. It is intrinsically linked to other key factors, including the LoRA rank ($r$), the learning rate ($\eta$), and the task-dependent gradient variance ($\sigma_L^2$). The final performance is a result of the complex interplay among all these components.

• **Interaction with Rank Size:** A key finding, highlighted in our ablation studies and consistent with our theoretical analysis, is that the benefits of magnitude scaling are not monotonic. We empirically observe that the performance gains from increased magnitude scaling tend to diminish and can even reverse at higher ranks. This is because larger ranks inherently amplify updates, as predicted by Proposition 2, suggesting that magnitude should be scaled more conservatively in high-rank settings.

• **On Optimality:** Our work aims to identify and validate update magnitude as a core mechanism influencing LoRA's training dynamics, thereby demystifying the success of methods like PiSSA. We do not claim to have identified an optimal magnitude scaling strategy. Determining the optimal magnitude for different models, tasks, and layers remains a challenging and important direction for future research.

In summary, the magnitude principle serves as a powerful analytical tool for understanding and designing LoRA-based methods, but its application should be contextualized within the broader optimization landscape.

