# OpenReview forum: "The Primacy of Magnitude in Low-Rank Adaptation"
_NeurIPS.cc/2025/Conference — NeurIPS 2025 spotlight_

### Official Review · Reviewer_62N4 · 2025-06-12

**Clarity:** 3
**Significance:** 3
**Originality:** 2
**Rating:** 5
**Confidence:** 4

**Summary:**

This paper introduces a unifying principle for understanding and improving Low-Rank Adaptation (LoRA), arguing that the magnitude of weight updates is the primary driver of its performance. The authors present a theoretical framework that connects learning rate, scaling factor, and initialization as different means to the same end: regulating update magnitude. Using this, the authors demystify the success of recent spectral initialization methods, positing that their effectiveness comes from amplifying update magnitudes, not from a more abstract preservation of pretrained knowledge.

Building on this insight, the paper proposes LoRAM that initializes the LoRA matrices using deterministic orthogonal bases derived from the Discrete Sine Transform and scales them by a factor carefully calculated to mimic the magnitude gains of spectral approaches. Through extensive experiments on a diverse set of language, coding, and vision-language tasks, the authors demonstrate that LoRAM consistently matches or outperforms more complex initialization schemes while retaining the full efficiency and plug-and-play simplicity of the original LoRA.

**Questions:**

No further questions

**Ethical Concerns:**

["NO or VERY MINOR ethics concerns only"]

**Final Justification:**

After reading other reviews which all go in the direction of acceptance, I maintain my score

**Limitations:**

The authors have adequately addressed the limitations of their work in a dedicated section. They correctly note that LoRAM mimics, rather than optimizes, spectral magnitudes and that further gains might be found by jointly optimizing magnitude with other hyperparameters like rank and learning rate.

**Paper Formatting Concerns:**

No formatting issues

**Quality:**

3

**Strengths And Weaknesses:**

Strengths:

This paper successfully challenges the findings of previous research by proposing an alternative explanation for the improved performance of spectral decomposition PEFT algorithms.

LoRAM displays comparable or slightly superior performance to spectral PEFT algorithms using a simpler strategy.

The findings are mathematically supported and insights into the training dynamics are presented in Figures 2 and 3.

LoRAM's performance is evaluated across a range of tasks to comfort as to it efficiency

Weaknesses:

(Minor) The ablation studies do show that while magnitude is the dominant factor, the choice of basis vectors (e.g., DST vs. SVD-derived) and the update direction (as influenced by data-driven methods like LoRA-GA) can still lead to minor performance differences outside of magnitude alone. The paper does acknowledge this.

---

> ### Author Rebuttal · Authors · 2025-07-30
>
> We sincerely thank Reviewer 62N4 for the thoughtful and constructive feedback. We are particularly grateful for the recognition of our work's **quality, clarity, and significance**, especially the acknowledgment that LoRAM **successfully** provides an alternative explanation for the improved performance of spectral decomposition PEFT algorithms. We also appreciate the recognition of the **mathematical support and insights** into training dynamics and the evaluation of LoRAM’s performance across various tasks.
>
> We concur with the reviewer’s observation that, while spectral value plays a key role, the choice of basis vectors and the update direction influenced by data-driven methods like LoRA-GA can indeed contribute to performance differences. We greatly appreciate the reviewer’s positive outlook on our contributions despite the nuances.
>
> We remain committed to incorporating feedback and engaging in ongoing improvements through future research.

---

### Official Review · Reviewer_Eyvp · 2025-06-21

**Clarity:** 3
**Significance:** 3
**Originality:** 4
**Rating:** 5
**Confidence:** 4

**Summary:**

This paper focuses on the parameter-efficient fine-tuning of large language models. The authors investigate the initialization methods of LoRA to improve its convergence and performance. Specifically, they theoretically re-interpret the success of spectral initialization in terms of weight magnitude and unify the hyperparameter for LoRA tuning as the magnitude regulation optimization. Furthermore, the authors propose an initialization method called LoRAM, which serves as a strong baseline for replacing spectral initialization across multiple benchmarks.

**Questions:**

According to the above weaknesses, I expect the authors to (i) demonstrate the efficiency of the proposed method, (ii) provide the reliance of their assumption.

**Ethical Concerns:**

["NO or VERY MINOR ethics concerns only"]

**Final Justification:**

The rebuttal of authors addresses my concerns about the LoRAM's memory usage and computation requirements. Thus, I have decided to maintain my positive rating.

**Limitations:**

Yes

**Quality:**

4

**Strengths And Weaknesses:**

**Strengths**:
1. This paper provides a comprehensive theoretical analysis of the correlation between the LoRA training dynamics and the weight magnitude.
2. This paper interprets the effectiveness of spectral initialization and unifies the hyperparameters of LoRA in terms of weight magnitude.
3. The proposed LoRAM is easy to implement and improves LoRA in multiple scenarios.
4. The proposed LoRAM is evaluated on multiple tasks and achieves comparable performance to spectral initialization methods.
5. This paper is well-organized and easy to follow.

**Weaknesses**:
1. The proposed LoRAM is designed to avoid the additional matrix decomposition and storage overhead of spectral initialization methods. Nevertheless, the current experiments only evaluate LoRAM's performance in comparison to spectral methods. Including comparisons of computational costs and memory usage would further demonstrate the effectiveness and efficiency of the proposed approach.
2. In the experiment of Figure 2, the pre-trained model and dataset should be described.
3. Since the authors assume that gradients $\nabla_W L^{(t)}$ follow a normal distribution in Proposition 2, it would be insightful to investigate whether this assumption holds reliably across different models, layers, and downstream tasks.
4. The LoRAM is designed for improving naive LoRA, which might lack the flexibility to accommodate the current practice of incorporating additional operations into LoRA [1, 2, 3].

[1] Expanding Sparse Tuning for Low Memory Usage, NeurIPS'24

[2] HydraLoRA: An Asymmetric LoRA Architecture for Efficient Fine-Tuning, NeurIPS'24

[3] Parameter-Efficient Fine-Tuning with Discrete Fourier Transform, ICML'24

---

> ### Author Rebuttal · Authors · 2025-07-30
>
> **General Response**
>
> We sincerely thank Reviewer Eyvp for the thoughtful and constructive feedback. We appreciate the positive evaluation of our work’s **quality, clarity, significance, and originality**. We are particularly grateful for the recognition of our strengths, including the **theoretical analysis** of LoRA training dynamics, the **unification of LoRA hyperparameters** in terms of weight magnitude, and the **practical contributions** of LoRAM in enhancing LoRA performance across tasks.
>
> We also value the reviewer’s constructive comments on areas for improvement and address them below.
>
> ___
>
> **Q1. Computational Cost and Memory Usage Comparison**
>
> We appreciate the reviewer's question regarding computational cost and memory usage, a crucial aspect for practical deployment.
>
> 1. **Theoretical Complexities** (Table 1): We provide the theoretical computational and memory complexities, where $m$ and $n$ represent the matrix dimensions, and $r$ represents the rank.
>
>    * LoRA: Its complexity primarily stems from random matrix initialization.
>    * LoRAM: Its complexity is driven by the initialization of the DST matrix, computed efficiently via vector multiplication (see provided code).  The magnitude gain factor $\beta$ can be computed explicitly as  $\nu[\Phi _ n\Phi _ m^\top]\approx r\nu[\Phi _ n]\nu[\Phi _ m^\top]=\frac{r}{mn}$ for the DST matrix.
>    * PiSSA: Its complexity is dominated by SVD, which requires significant storage for the initial state and result matrices, leading to a space complexity of $O(2r(m+n))$.
>
> Table 1. Theoretical initialization time and space complexities of LoRA, LoRAM, and PiSSA.
>
> |       | LoRA        | LoRAM       | PiSSA                 |
> |-------|-------------|-------------|-----------------------|
> | Time  | $O(mr+nr)$    | $O(mr+nr)$    | $O(\min(m^2n, mn^2))$ |
> | Space | $O(r(m+n))$ | $O(r(m+n))$ | $O(2r(m+n))$          |
>
>
> **2. Practical Performance** (Table 2): We tested training LLAMA-7B on an 8-GPU setup. LoRA and LoRAM exhibited similar practical performance. PiSSA has two implementations:
>
> * **Official Default Pre-processing**: Saves the residual model and LoRA results, leading to high space usage. This is the recommended approach.
> * **Direct Transformers and PEFT Workflow**: Computes the SVD matrix on the CPU before transferring it to the GPU (due to discrepancies with fast SVD across different GPU devices). This results in significantly slower performance, consistent with **known bottlenecks mentioned in PiSSA’s official GitHub issues**. In our 8-GPU setup with Deepspeed distribution training, the direct process typically took over half an hour. Since we cannot confirm whether the time was accurately measured, we report only the approximate duration.
>
>
> Table 2. Practical initialization time and space cost of LoRA, LoRAM, and PiSSA.
>
>
> |               | LoRA  | LoRAM | PiSSA (pre-process) | PiSSA (direct) |
> |---------------|-------|-------|---------------------|----------------|
> | Time ($r=16$) | 1.36s | 0.95s | 51.32s              | $>$ 10min      |
> | Time ($r=128$)  | 4.23s | 2.19s | 57.79s              | $>$ 10min      |
> | Time ($r=512$)  | 8.50s | 5.19s | 110.31s             | $>$ 10min      |
> | Space ($r=16$)  | 152MB | 152MB | 12.5GB  + 152MB     | 305MB          |
> | Space ($r=128$) | 1.2GB | 1.2GB | 12.5GB + 1.2GB      | 2.4GB          |
> | Space ($r=512$) | 4.8GB | 4.8GB | 12.5GB + 4.8GB      | 9.6GB          |
>
>
> Code 1. A Pytorch Code for Generating DST matrix
>
> ```angular2html
> import torch
>
> def fast_dst_matrix(m, n, device='cpu', dtype=torch.float32):
>     """
>     generate m x n DST-I matrix.
>     """
>     transpose = False
>     if m < n:
>         m, n = n, m
>         transpose = True
>
>     assert n <= m, "n must be smaller than m"
>
>     k = torch.arange(n, device=device, dtype=dtype).unsqueeze(1)  # shape (n,1)
>     i = torch.arange(m, device=device, dtype=dtype).unsqueeze(0)  # shape (1,m)
>
>     dst_basis = torch.sin((i + 1) * (k + 1) * torch.pi / (m + 1))
>
>     scale = torch.sqrt(torch.tensor(2 / (m + 1), device=device, dtype=dtype))
>
>     dst_matrix = (scale * dst_basis).T
>
>     if transpose:
>         dst_matrix = dst_matrix.T
>
>     return dst_matrix.to(device=device)
> ```
>
> ---
> **Q2.Model and Dataset Description in Figure 2**
>
>
> Thank you for pointing this out.
>
> As mentioned in lines 100-103, we conducted a controlled experiment using a 5-layer toy MLP with an intermediate dimension of 400, where the model weights were initialized randomly. The data were generated from random noise (both random input and random output). Specifically, we fixed the random model weights and trained the LoRA module to fit the data mapping from scratch. This setup is simple but effectively tests LoRA's fitting ability. We will add more details in the revision.
>
>
> ---
> **Q3. Assumption of Normal Gradient Distribution in Proposition 2**
>
> Thank you for this insightful question. We would like to prove this reliability from three perspectives:
>
> 1. **Relaxed to zero-mean and independence**.
>    We state the critical properties required for our derivation are that the gradient entries are zero-mean and independent to simplify the mathematical derivation. While the Gaussian assumption is a standard and convenient way to ensure these properties, it is not strictly necessary.
> 2. **Empirical Validation**.
>    We conducted an empirical investigation using LLaMA-2 for three NLG tasks and FLUX.1-12B for image customization. We monitored and visualized weight gradients across various layers. Our findings are:
>    * The gradient entries consistently have zero mean, sampled from various layers and at different training stages.
>    * The gradient distributions are Gaussian-shaped, with a single peak, while the variance varies across layers and   remains stable in the middle of training.
> 3. **Prior Works**.
>    This assumption is well-justified and widely adopted in the literature. For example:
>    * \[1] provides extensive illustrations of normal gradient distributions.
>    * \[2] makes assumption of zero-mean Gaussian distribution throughout the article, inspiring us to adopt this assumption in our work.
>    * \[3] presents a great work that derives the mean and variance of forward and backward signals in transformer architectures.
>
> ---
>
> **Q4. Flexibility of LoRAM in Incorporating Additional Operations**
>
> We appreciate the reviewer’s insightful point.
>
> Indeed, as you rightly pointed out, LoRAM is designed to improve the original LoRA method and may lack the flexibility to incorporate the great works you mentioned, which are orthogonal to LoRAM's core idea.
>
> Although LoRAM does not directly integrate with these methods, we consider that our theoretical analysis could offer some insights into understanding these approaches, making it a good direction for future research.
>
> ---
>
> \[1] Understanding the difficulty of training deep feedforward neural networks.  In AISTATS, 2010.
>
> \[2] Delving deep into rectifiers: Surpassing human-level performance on imagenet classification. In ICCV, 2015.
>
> \[3] Transformers get stable: An end-to-end signal propagation theory for language models. In ICML, 2024.

---

> > ### Comment · Reviewer_Eyvp · 2025-08-05
> >
> > Thank the authors for their detailed rebuttal, which addresses my concerns. Thus, I have decided to maintain my original rating. Furthermore, I expect the authors to explore and discuss the potential extension of LoRAM to improve its compatibility with existing nonlinear approaches in their future work. Because overcoming the linear constraints (i.e., the low-rank property of the $\Delta W$) may offer a fundamentally solution for approximating full fine-tuning with low-rank matrices. If LoRAM can inspire or lead to such developments, it would be great to the research community.

---

> ### Author Response · Authors · 2025-08-05
> **Thanks for Your Response and Support**
>
> Dear Reviewer,
>
> Thank you very much for your positive response and continued support of our work. We deeply appreciate your thoughtful suggestion. We fully agree with your perspective, and your comment has been truly inspiring.
>
> We will mention this insightful suggestion in the revised paper. The idea of extending LoRAM to improve compatibility with nonlinear approaches is indeed a promising direction. We are particularly interested in exploring whether the magnitude principle can be applied in nonlinear settings, as this could provide an essential foundation for approximating full fine-tuning with low-rank matrices.
>
> Once again, thank you for your valuable feedback and constructive suggestions.
>
> Sincerely,
>
> The Authors of Submission 4604

---

### Official Review · Reviewer_eWC2 · 2025-07-01

**Clarity:** 1
**Significance:** 3
**Originality:** 3
**Rating:** 4
**Confidence:** 3

**Summary:**

This paper investigates the important problems of training dynamics and initialization in Low-Rank Adapters (LoRA). The authors develop an initialization (LORAM) based on deterministic orthogonal bases with controllable magnitude scaling parameters. This developed theoretically by treating the evolution of LoRA parameter magnitudes as a linear dynamical system and observing the magnitude changes during training. A correspondence between different optimization schemes is developed based on a scaling factor applied to adapters.

The authors demonstrate that the LORAM initialization achieves competitive performance to leading LoRA schemes across a range of Natural Language Generation, Natural Language Understanding, and Vision-Language tasks with superior performance achieved in several individual benchmarks. They additionally provide a number of additional analyses and ablations for their method, including the use of a 'tracking mode' that backsolves scaling parameters for LORAM based on target initialization schemes.

**Questions:**

My current view is that this is a decent paper which presents interesting theoretical analysis along with well benchmarked and ablated experiments, but which would benefit substantially from improving clarity (especially for figures). In the current form many of the details are not immediately clear which detracts from the analysis and results. My suggestion would be that simplifying figures and sections by splitting less essential details to the supplementary materials could make this a good paper.

In addition to the details raised in the Strengths and Weaknesses, there are a few questions which it would be useful to clarify:

* Could the authors clarify Figure 2a and 2b? As described in the Strengths and Weaknesses section these figures are difficult to digest (e.g. the meaning of the SGD region and the Adam region, and the interpretation of the loss and norm differences). My interpretation of b) is that it shows how changing the initial variance of the A and B matrices changes the total magnitude across training, but I am not certain
* Could the authors clarify Figure 3? It appears to either show how the spectral concentration and gain factor change over training or across the dimension of the layers? In addition the insert in the p[r] row seems to refer to the early Q[r] values.
* There are a few numbers that are used in the paper (e.g. L126-130, sigma_A = 1/20, k1 = 1/16) where it isn't immediately clear how these were selected and the how it connects with the theoretical analysis
* In the Abstract the method is described as a `Basis and Basis' initialization scheme, however this isn't clearly defined later
* The linear dynamical system approach used within Proposition 2 is an interesting and potentially novel way to analyze LoRA. As section 2.3 and the supplementary proof don't directly include citations, could the authors briefly discuss the papers they feel most relevant to the approach used in Proposition 2.

**Ethical Concerns:**

["NO or VERY MINOR ethics concerns only"]

**Final Justification:**

While I have remaining concerns on the clarity of the figures / technical details I have raised my review to a borderline accept. The reasons for this are the technical contribution of the paper (which I believe is interesting and relevant), the depth of experimentation, and the author's detailed responses to all reviewers during the review process in which they have sincerely worked to clarify and validate their claims. The authors have committed to improving the clarity of the manuscript and figures for any camera-ready version. I wish them the best on this project and thank them for their engaged and responsive discussion.

**Limitations:**

Yes.

**Paper Formatting Concerns:**

The most significant formatting issue relates to figures (legibility).

**Quality:**

2

**Strengths And Weaknesses:**

The paper is well motivated. Despite the many variants of LoRA there are still many open questions to its training dynamics and optimal initialization. The paper approaches this in an ambitious way: by seeking to develop a correspondence between the training dynamics under different optimization schemes and adapter magnitudes, the authors aim to unify understanding for a number of hyperparameter and design choices involved in LoRA.

The main strength of this paper is in its theoretical insight. In particular, the linear dynamical system approach for LoRA is interesting. In addition, the use of a deterministic orthogonal initialization (in particular the Discrete Sine Transform) is a very interesting approach which may motivate further exploration into other deterministic bases.

The experimentation and ablations are thorough and appropriate. Experiments appear to support the central idea that appropriately scaling parameters can boost LoRA convergence, and results show competitive or improved performance on a wide range of benchmarks across modalities (NLG, NLU, Vision-Language).

In terms of written description, the manuscript is relatively clear - with the related literature presented especially well.

There are a few important areas where the clarity is reduced: in particular, the figures are very information dense (for example, Figure 2a attempts to combine insights into multiple [a', a_A, a_B] hyperparameter ablation, optimizer [SGD, Adam], losses, and norm differences into a single double-axis figure with overlapping convergence lines [which I believe is meant to indicate they achieve equivalences as per Proposition 1]). This makes the figures very difficult to parse as a reader. Figure font sizes are often much too small to be visible when printed and when zoomed in. Adding in labels for axes (e.g. on Figures 2, 3, 5) would similarly improve the figures.

Tables 1, 2, and 3 are presented clearly and show that LORAM is competitive with leading methods, and achieves superior performance for several tests. Presentation for Table 4 is relatively dense (showing ablations for different scaling factors, bases, benchmark initialization methods, backsolved `tracking' initialization, etc) and could benefit from being split into separate tables in the supplementary materials to improve clarity. Bold figures could help make this comparison and the takeaway message clearer.

My general impression is that this paper has a lot of information that it wants to convey in a short amount of space. This unfortunately makes it relatively difficult to digest as a reader. Spacing these ideas out by focusing on the key message / simplified clear figures in the main paper, and moving less critical ablations to the supplementary materials would greatly improve this paper's potential impact. (E.g. - separating a few of the ablations in Figures 2 and 3, and providing sufficient space to describe this in the supplementary).

Similar issues appear in some of the mathematical descriptions (very densely written, with some overloading of notation [e.g. it is not clear how B^{(0)} differs from B_{init}], and many details feeling a little "cramped" to fit in the full technical details of ablations, e.g. logarithmic gain factor, tracking mode, and the DST basis). Including many technical components in the main paper can mean that there is less space available to develop, motivate, and connect each idea clearly. In contrast, I found the proofs in Supplementary B were relatively clear and well developed. Focusing in on the most important message (things that directly support the LORAM initialization and magnitude analysis), and leaving fuller description in the supplementary would make it much easier to digest (e.g. details of the LORAM initialization are provided relatively late in the method on page 6, and felt difficult to directly connect to the earlier formal linear dynamical system analysis and update magnitude scaling).

---

> ### Author Rebuttal · Authors · 2025-07-30
>
> **General Response**
>
> We sincerely thank Reviewer eWC2 for the thoughtful and constructive feedback. We are especially grateful for the recognition of our contributions across **theory, methodology, and experiments**, including the linear dynamical analysis, magnitude-based perspective, use of deterministic bases (e.g., DST), and extensive empirical validation.
>
> While we appreciate the description of our work as "decent," we understand that the main concerns lie in presentation clarity, rather than the technical soundness:
>
> 1. **Visual Density**: We acknowledge that some figures and tables are dense, which resulted from our attempt to present comprehensive findings within the page constraints.
> 2. **Information Overload**: With the large number of original contributions, we focused on precision, particularly in the theoretical sections, which may have compromised simplicity.
> 3. **Structural Suggestions**: We value the reviewer’s structural suggestions and agree that more space is needed to clearly develop, motivate, and connect each idea. We will revise the organization for greater clarity.
>
> We regret any confusion and respectfully ask for your understanding. With the opportunity to revise (e.g., in a camera-ready version with an additional page), we are confident we can present the content more clearly.
>
> ---
> **Q1. Confusion about Notation $B^{(0)}$ and $B_{\text{init}}$**
>
> We appreciate the reviewer’s careful review.  $B^{(0)}$ is used as a **general notation** to represent the initial state of the LoRA matrix $B$ at the training step $0$,  while $B_{\text{init}}$ refers to the **specific assignment** based on the chosen initialization method, which we specify more precisely as follows:
>
> * **In Proposition 1**, $B_{\text{init}}$ refers to any matrix of the appropriate shape for generality.
> * $B_{\text{init}}$ could also take values from three specific schemes:
>
>   1. Zeros: Used in the "Noise & Zeros" initialization of conventional LoRA.
>   2. $B_{\text{SVD}}$ (Eq. 7): Denotes the SVD-based initialization, as used in methods like PiSSA.
>   3. $B_{\text{LoRAM}}$ (Eq. 11): Represents the initialization using Discrete Sine Transform (DST) bases, as proposed in our LoRAM scheme.
>
> ---
> **Q2. Connection between Update Magnitude Scaling (Sec 2.2), Dynamical System Analysis (Sec 2.3), and LoRAM Initialization (Sec 3)**
>
> We establish the following logical progression:
>
> 1. Section 2.2 explains "Why **weight update magnitude** ($\nu[\Delta W _ {\text{LoRA}}^{(t)}]$) matters," showing that various hyperparameters converge to a common effect: they control the update magnitude, which directly influences loss reduction.
> 2. Section 2.3 demonstrates "How **initial variance** ($\nu[A^{(0)}]$ and $\nu[B^{(0)}]$) explicitly affect **weight update magnitude** ($\nu[\Delta W _ {\text{LoRA}}^{(t)}]$)."
> 3. Section 3 illustrates "How **spectral initialization** and **LoRAM initialization** influence **initial variance** ($\nu[A^{(0)}]$ and $\nu[B^{(0)}]$), and in turn, affect **weight update magnitude** ($\nu[\Delta W _ {\text{LoRA}}^{(t)}]$)."
>
>
> ---
> **Q3. Clarification of Figure 2**
>
> These figures illustrate the training dynamics of low-rank adaptation under different initialization settings and optimizers, aiming to provide an intuitive validation and interpretation for the training dynamics and magnitude evolution.
>
> 1. **Figure 2a**: This figure shows the training dynamics of LoRA under two optimizers: SGD (for the first 2.5k steps) and Adam (thereafter). This setup validates Proposition 1, which states that when the product $\alpha' \alpha_A \alpha_B$ is held constant, the training trajectories of Adam and SGD should be equivalent. This equivalence is reflected in the overlapping loss curves (black, blue, green, red). To further verify that $\Delta W _ {\text{LoRA}}^{(t)} = \Delta \tilde{W} _ {\text{LoRA}}^{(t)}$ across training dynamics, we compute the norm difference $||\Delta \tilde{W} _ {\text{LoRA}}^{(t)} - \Delta W _ {\text{LoRA}}^{(t)} ||_F^2$ between the weights learned by the two optimizers and the baseline weights (black curve), accumulated across all layers. This confirms that not only the loss, but also the learned weights, evolve similarly across settings.
>
> 2. **Figure 2b**: Your interpretation is correct. This figure shows how varying the initial variance of the $A$ and $B$ matrices affects the update magnitude $\nu[\Delta W]$ during training. As shown, larger initialization variance leads to larger parameter updates and faster convergence, which supports our core claims in Proposition 2. Additionally, as noted in Eq.(5), the magnitudes of $A$ and $B$ remain relatively stable throughout training, further emphasizing the importance of their initial values.
>
> ---
> **Q4. Clarification of Figure 3**
>
> Figure 3 illustrates how spectral concentration ($\rho[r]$) and the spectral gain factor ($Q[r]$) vary across different rank configurations (x-axis) and model layers (as shown in the legend).
>
> 1. The plot of $\rho[r]$ shows that $\rho[r]$ decreases as rank $r$ increases. According to Equation (9), this indicates that top principal components contribute most significantly to initialization magnitude and optimization dynamics.
>
> 2. The plot of $Q[r]$ generally suggests that as rank increases, magnitude scaling becomes more significant, with higher-rank configurations leading to larger overall update magnitudes. For the "Noise and Zeros" initialization, $k_1 = r(m\frac{1}{m^2}+0) = \frac{r}{m}$, resulting in constant growth (represented by the white line). In contrast, PiSSA produces diverse curves (colored lines) due to varying singular values across models and layers. We approximate the spectral gain factor using a logarithmic function (shown for three cases) to moderate the enhancement of magnitude scaling.
>
> 3. The inset plot highlights low-rank configurations, which are most commonly used in practice. We will add arrows or labels to make this relationship clearer.
>
> ---
> **Q5. Clarification of Numbers (e.g., $\sigma_A = 1/20$, $k_1 = 1/16$)**
>
> We apologize for not providing a clearer derivation of these values in the main text.
>
> 1. **For $\sigma_A = 1/20$**: This value is based on the Kaiming initialization principle. As described in lines 100-102, we conducted a controlled experiment using a 5-layer MLP with an intermediate dimension of 400. According to Kaiming initialization, the weight variance is set to $1/\sqrt{n}$ (where $n$ is the input size) to maintain proper variance. In this case, $\sigma_A = 1/\sqrt{400} = 1/20$.
> 2. **For $k_1 = 1/16$**: This value is derived based on the experimental setup where the LoRA rank $r = 25$. Taking into account the chosen rank and intermediate dimension, $k_1$ is calculated as: $ k_1 = r(m\sigma_A^4 + n\sigma_B^4) = 25 \left( 400 \times \frac{1}{20^4} + 400 \times 0^4 \right) = 1/16$
>
>
>
> ---
> **Q6. Clarification of "Basis and Basis" Initialization Scheme**
>
> As illustrated in Figure 1 and Algorithm 1 of our paper, our method employs a deterministic orthogonal **basis** (e.g., DST) for each LoRA matrix. This approach contrasts with the conventional "Noise & Zeros" initialization, where matrix $A$ is typically initialized with random noise, and matrix $B$ with zeros.
>
> ---
> **Q7. Papers relevant to the approach used in Proposition 2**
>
> We sincerely thank the reviewer for their positive assessment of our approach as **"interesting and potentially novel."** Our thinking was informed by a broader line of research that models the dynamics of neural network training. We were particularly inspired by the general philosophy in work:
>
> 1. **Gradient Flow :** Which models parameter training trajectories as solutions to differential equations on a loss landscape [1].
> 2. **Neural ODEs :** Which provide a powerful framework for thinking about deep networks as continuous-time dynamical systems [2].
>
>  The modeling of linear dynamical systems is novel and based on the unique update structure of LoRA:
> * The gradient for the matrix $A$: $\nabla_A L= B^\top (\nabla_W L)$ is a function of the current state of $B$.
> * Conversely, the gradient for $B$: $\nabla_B L = (\nabla_W L) A^\top$ is a function of the current state of $A$.
>
> The optimization creates a feedback loop that can be expressed conceptually as $A_{t+1} = A_t + f(B_t)$ and $B_{t+1} = B_t + g(A_t)$, naturally simplified into the linear dynamical system. The remaining analysis is based on our unique efforts.
>
> ---
> [1] Gradient flows: in metric spaces and in the space of probability measures. In Basel: Birkhäuser Basel. 2005
>
> [2] Neural ordinary differential equations. In NeurIPS 2018

---

> > ### Comment · Reviewer_eWC2 · 2025-08-03
> >
> > I thank the authors for their detailed rebuttal and clarifications. In particular providing details on the motivating literature for the linear dynamical systems analysis, paper terminology (e.g. Basis & Basis initialization), and providing further explanation on figures.
> >
> > I admit that I still find the figures difficult to digest. For example, it isn't immediately clear why SGD is presented for 2.5k steps and then Adam (this gives what I assume to be an incorrect impression that different optimizers were used in these two regions). I believe that in Figure 3, the goal is to show that 1) the spectral energy is highly concentrated, and 2) magnitude scaling can be approximated by a logarithmic factor of rank. The figure doesn't make explicit reference to PiSSA so I'm unsure how it is related.
> >
> > While I believe this work does present an interesting finding, the overall issues of clarity are unfortunately significant enough in my eyes to require a substantial revision of figures and rewriting. I am not sure this can be adequately achieved without significant edits to a camera ready version (acknowledging the extra 1 page allowed material).
> >
> > As a result I'm maintaining a Borderline Reject rating. I note that reviewers Eyvp and 62N4 are more positive on the current version and I would not object if consensus ultimately favors acceptance.

---

> > > ### Author Response · Authors · 2025-08-05
> > > **Clarification Follow-Up**
> > >
> > > Dear Reviewer eWC2,
> > >
> > > Thank you once again for your thoughtful comments.
> > >
> > > We would like to kindly follow up to ensure that if our previous response has helped clarify the intent and construction of Figures 2 and 3. Your continued feedback will be valuable in improving the quality of our paper.
> > >
> > > If there are any remaining points of confusion or further suggestions, we would greatly appreciate the opportunity to engage in additional discussions to address any remaining questions.
> > >
> > > Best regards,
> > >
> > > The Authors of Submission 4604

---

> > > > ### Comment · Reviewer_eWC2 · 2025-08-05
> > > >
> > > > While I have remaining concerns on the clarity of the figures / technical details I have raised my review to a borderline accept.
> > > >
> > > > The reasons for this are the technical contribution of the paper (which I believe is interesting and relevant), the depth of experimentation, and the author's detailed responses to all reviewers during the review process in which they have sincerely worked to clarify and validate their claims.
> > > >
> > > > With that said I believe that the paper should undergo a substantial revision for a camera-ready submission if the paper is accepted. I acknowledge the authors have sought to clarify the details for figures during the rebuttal - and believe that they have expressed the intent behind these figures more clearly during the rebuttal. In the current version of the paper this information cannot be directly inferred from the figures and manuscript text (even with detailed explanation).
> > > >
> > > > As a result acceptance should involve (at a minimum) replacing Figures 2, 3, and 5 to present the desired information in a clearer way. This may require breaking the figures into separate graphs, and placing some details in the Supplementary Materials. Simplified figures that illustrate the key message (rather than crowding in a large number of hyperparameter / experiment ablations) would make for a large improvement of clarity over the current version.
> > > >
> > > > Axes, labels, line weights, font size changes, and in text description for figures would be required for legibility and to allow the explanations provided in the rebuttal to be reflected in the manuscript. Figure 2a) in particular should be alternatively expressed (without the current Adam / SGD region presentation). Minor changes to the captioning or labels would unfortunately not be enough to make these details clear.
> > > >
> > > > I thank the authors for their detailed engagement during the rebuttal process and wish them the best for this project.

---

> > > > > ### Author Response · Authors · 2025-08-05
> > > > > **Thanks for Your Response and Support**
> > > > >
> > > > > Dear Reviewer eWC2,
> > > > >
> > > > > Thank you very much for your thoughtful reconsideration and for raising your review to a positive score. We are truly grateful for your recognition of the paper’s technical contributions, experimental depth, and our sincere engagement during the review process.
> > > > >
> > > > > We also greatly appreciate the clear roadmap you provided. In the revised version, we are working to enhance the clarity of the key figures and improve the overall presentation. Additionally, we will ensure that the information is presented more effectively, potentially moving some content to the supplementary materials.
> > > > >
> > > > > We acknowledge that this exchange has been very beneficial to us, not only in improving this paper, but also in deepening our understanding of concise expression.
> > > > >
> > > > > Thank you again for your time, constructive feedback, and support.
> > > > >
> > > > > Best regards,
> > > > >
> > > > > The Authors of Submission 4604

---

> ### Author Response · Authors · 2025-08-03
> **Response to Reviewer eWC2's Comment**
>
> **Dear Reviewer eWC2,**
>
> We sincerely appreciate your thoughtful follow-up and your detailed feedback. We especially want to thank you for this note:
>
> > “I would not object if consensus ultimately favors acceptance.”
>
> Your openness to the group's consensus means a great deal to us. We would also like to offer a brief clarification on the two figures you mentioned, as we believe it might resolve the remaining confusion.
>
> ---
>
> ### **Clarification on Figure 2: Use of Two Optimizers**
>
> You are correct that different optimizers were used in the two regions of the curve. Specifically, we deliberately used **SGD for the first 2.5k steps** and **Adam for the following 2.5k steps** to empirically verify **Proposition 1**, which shows that when the product \$\alpha' \alpha\_A \alpha\_B\$ is held constant, **LoRA exhibits equivalent training trajectories under common optimizers** (e.g., SGD and Adam). Thus, Figure 2 is designed to demonstrate this equivalence in practice, with the overlapping loss and weight update curves confirming our theoretical claim.
>
> Importantly, the switch in optimizer and the specific number of steps do **not** influence the conclusion. The key result—that **equivalent update magnitude leads to equivalent training behavior—holds as long as the product $\alpha' \alpha_A \alpha_B$ remains constant, regardless of the optimizer or scheduling choices.**
>
> To enhance clarity, we will include more detailed illustrations in the appendix of the revised version—presenting separate figures for SGD and Adam.
>
> ---
>
> ### **Clarification on Figure 3: Its Direct Relationship to PiSSA**
>
> You are correct in your interpretation. The figure's goal is to visualize the magnitude gain that methods like **PiSSA** leverage. To clarify its direct link:
>
> - The solid colored curves, $\rho[r]$ (spectral concentration factor) and $Q[r]$ (spectral gain factor), are calculated directly from the SVD of pretrained weights—the mechanism used by **PiSSA**. Therefore, these curves represent the magnitude dynamics of a **PiSSA initialization**.
>
> - The plot illustrates two points from our analysis of PiSSA:
>     - $\rho[r]$ shows that spectral energy is highly concentrated in the top singular values, explaining why PiSSA is effective especially at low-rank settings.
>     - $Q[r]$ quantifies how PiSSA's initialization (colored lines) achieves a significantly larger magnitude gain compared to the naive "Noise & Zeros" LoRA baseline (the white dotted line). This is the gain that LoRAM successfully mimics with a simpler approach.
>
> We will revise the caption to make this connection more explicit.
>
> ---
>
> We appreciate the reviewer’s note that **Reviewer Eyvp and Reviewer 62N4 have provided positive feedback** regarding this aspect, while we also understand the reviewer’s concern about clarity and recognize that reviewers may have varying perspectives on paper’s clarity due to differing backgrounds. Given the policy limitations during the rebuttal phase, we regret that we cannot provide an updated version of the figures and manuscript at this stage. However, we are committed to refining the structure and wording in the final version to ensure clearer communication of our research.
>
> Lastly, we are encouraged that you found our work to contain an **"interesting finding"**, and we sincerely thank you once again for your valuable feedback and thoughtful engagement. **We sincerely hope that our clarifications help alleviate most of your concerns and contribute to a more favorable reassessment of our work.**
>
>
> ---
>
> Sincerely,
> The Authors of Submission 4604

---

### Official Review · Reviewer_PbY3 · 2025-07-03

**Clarity:** 2
**Significance:** 2
**Originality:** 2
**Rating:** 4
**Confidence:** 3

**Summary:**

LoRAM introduces a magnitude-aware initialization scheme for LoRA that replaces SVD with deterministic orthogonal bases scaled by pretrained weight statistics. While preserving the standard LoRA training pipeline, this approach achieves comparable or superior performance with significantly reduced computational overhead. By reinterpreting spectral methods through the lens of update magnitude, the paper provides a unifying perspective on various hyperparameter strategies and highlights magnitude scaling as a key lever for efficient adaptation.

**Questions:**

Please refer to weakness

**Ethical Concerns:**

["NO or VERY MINOR ethics concerns only"]

**Final Justification:**

While the theoretical motivation and related issues have been adequately addressed in the rebuttal, I remain unconvinced that the underperformance of MiLoRA and other baselines has been fully explained. Although the authors have discussed possible causes—such as differences in optimal learning rates—it is still unclear whether these baselines were appropriately tuned for each task. The appendix does not provide sufficient detail to verify this. As a result, this concern remains unresolved.

**Limitations:**

Please refer to weakness

**Quality:**

2

**Strengths And Weaknesses:**

(+) Provides a unified theoretical framework linking LoRA hyperparameters through update magnitude. This perspective allows for a coherent interpretation of prior LoRA variants such as RsLoRA and LoRA+, which represents a clear theoretical contribution.

(+) The ablation studies are appropriately designed and help isolate the effects of key components in the proposed method.

(-) The authors reinterpret SVD-based initialization—commonly understood as preserving principal directions—as primarily contributing to performance through magnitude amplification. This is an insightful and valid perspective. However, it is important not to dismiss the spatial or spectral interpretations of SVD entirely. The original LoRA paper analyzes the similarity between $W$ and $\Delta W$ in weight space, and later methods such as MiLoRA and CorDA explicitly leverage the structure of principal and minor components to mitigate forgetting and enhance transferability. Notably, MiLoRA achieves strong performance even when modifying only small singular components, making its underperformance in this paper somewhat surprising and potentially in need of further clarification.

(-) The overall narrative may slightly overstate the universality of the magnitude principle. In practice, the benefit of increasing update magnitude likely depends on factors such as rank size, model architecture, and task characteristics. A more nuanced discussion of when magnitude scaling is helpful—and when it may result in diminishing returns or instability—would strengthen the overall framing.

(-) As shown in Figure 5(a), the improvement in convergence speed may not be as pronounced as implied, and appears moderate relative to other methods.

(-) While LoRAM achieves consistent performance gains in Tables 1 and 2, the improvements over strong baselines are relatively modest. The results are solid and reliable, but in some cases the margin may be too small to be considered practically significant.

---

> ### Author Rebuttal · Authors · 2025-07-30
>
> **General Response**
>
> We sincerely thank Reviewer PbY3 for the thoughtful and constructive feedback. We greatly appreciate your recognition of the strengths of our work, including the **insightful theoretical framework, the appropriate experimental design and the solid performance**.
>
> Regarding the concerns raised, we categorize them into two types of issues:
> 1. **Differences in Interpretation**: We feel that concerns on the dismissal of spectral interpretation (Q1) and the overstatement of the magnitude principle (Q3), may arise from differences in interpretation of our work. We hope to further clarify these points to mitigate concerns.
> 2. **Method Performance**: Concerns regarding findings such as the underperformance of MiLoRA (Q2) and the modest improvements (Q4) are valid points that we believe are addressable through further explanation.
>
> We apologize for any confusion caused by points we intended to convey, but which may not have been sufficiently clear. We will clarify and address these issues in the revision.
>
> ---
> **Q1. Concern on Dismissal of Spectral Interpretations**
>
> We agree that spectral interpretation is significant. In fact, we do not dismiss the spectral view, but rather provide a deeper and mechanistic understanding of its effectiveness through the lens of the magnitude principle.
>
> 1. **Relationship to prior spectral analyses**. The original LoRA paper [1] (Sec 7.3) observed that the adapted weight $\Delta W$ hugely amplify features absent in $W$. **Our work extends this by explicitly modeling the training dynamics behind this amplification**, further revealing how LoRA's inherent low-rank structure can weaken this expected effect, and crucially, how subsequent hyperparameter tuning further amplify update magnitudes to enhance performance.
>    * **Spectral values**. Our findings suggest that in most spectral initialization schemes, the major performance gains are primarily driven by the magnitude amplification resulting from scaling the singular values.
>    * **Spectral bases**. As shown in lines 257-259, we also acknowledge that the spectral matrices from full gradients (in LoRA-GA) have significant impacts. We prove its effectiveness stems from maximizing gradient magnitudes. This directly aligns with our magnitude principle. We will highlight this finding in Sec 2.3.
> 2. **Relationship to Minor-component methods**. Methods like MiLoRA and CorDA tune minor components to preserve key information and mitigate forgetting, while there still lacks a theoretical framework to explain their effectiveness. **Our study, within the scope of enhancing convergence and performance, showcases that leveraging minor components typically yield less magnitude amplification.** This suggests their benefits in avoiding overfitting and providing a more stable solution might stem from this inherent lower magnitude scaling.
>
> ---
> **Q2. About the underperformance of MiLoRA**
>
> In summary, PiSSA generally outperforms MiLoRA at typical fine-tuning learning rates (e.g., 2e-5), though this advantage diminishes or reverses with larger learning rates.
>
> **Intuitive explanation:** MiLoRA directly uses the smallest singular values and bases, which, being close to zero, can induce a vanishing problem.
>
> **Practical guidance:** Our theory predicts that PiSSA's stronger magnitude amplification from principal singular values necessitates a smaller optimal learning rate compared to MiLoRA. We demonstrate this as follows:
>
> 1.  **Consistent Findings with PiSSA**. Our experiments, conducted at a moderate learning rate (2e-5), align with the PiSSA paper's setups and findings that "principal components consistently lead to reduced training loss and enhanced accuracy" (Figure 8, PiSSA paper [2]).
> 2.  **Experiments with Large Learning Rates**. To better understand MiLoRA's comparative performance, we conducted experiments at a higher learning rate (2e-4), matching that used in the MiLoRA paper. The tables below reveal several interesting patterns:
>     1.  **At a lower rank (r = 16)**, all methods benefit from the increased learning rate (2e-5 to 2e-4), showing improved performance across tasks, indicating higher learning rates can enhance convergence in low-rank settings.
>     2.  **At a higher rank (r = 128)**, a divergence emerges. While most datasets still show improvement, PiSSA exhibits a notable performance drop on the Commonsense dataset, falling below its rank-16 performance and MiLoRA's.
>     3.  **Training curves**. PiSSA's loss plateaus early and fails to decrease at high-rank, high-LR configurations, suggesting amplified updates overshoot the effective descent step, preventing loss reduction.
>
> These findings are **fully consistent with our claims in Section 4.3 (lines 262–263)**: "Magnitude scaling should be applied conservatively at higher ranks, since larger ranks inherently amplify updates."
>
>
>
>
> Table 1: Results of PiSSA, LoRAM and MiLoRA with different learning rates ($r = 16$).
>
> |               | PiSSA | LoRAM | MiLoRA | PiSSA | LoRAM | MiLoRA |
> |---------------|-------|-------|--------|-------|-------|--------|
> | Learning rate | 2e-5  | 2e-5  | 2e-5   | 2e-4  | 2e-4  | 2e-4   |
> | GSM8K         | 37.68 | 40.32 | 29.70  | 52.46 | 53.29 | 46.62  |
> | MATH          | 5.16  | 5.30  | 4.18   | 8.80  | 8.92  | 6.18   |
> | HumanEval     | 18.37 | 18.92 | 14.69  | 24.47 | 25.62 | 17.14  |
> | MBPP          | 28.62 | 28.83 | 27.23  | 31.22 | 31.74 | 28.65  |
> | Commonsense   | 73.72 | 75.19 | 67.90  | 77.02 | 79.11 | 76.67  |
>
> Table 2: Results of PiSSA, LoRAM and MiLoRA with different learning rates ($r = 128$) .
>
> |               | PiSSA | LoRAM | MiLoRA | PiSSA | LoRAM | MiLoRA |
> |---------------|-------|-------|--------|-------|-------|--------|
> | Learning rate | 2e-5  | 2e-5  | 2e-5   | 2e-4  | 2e-4  | 2e-4   |
> | GSM8K         | 51.48 | 51.12 | 39.81  | 58.37 | 59.28 | 54.66  |
> | MATH          | 7.04  | 7.25  | 5.18   | 11.46 | 10.76 | 9.20   |
> | HumanEval     | 21.62 | 22.03 | 20.39  | 26.81 | 29.95 | 28.02  |
> | MBPP          | 31.07 | 31.53 | 29.95  | 36.54 | 37.80 | 33.35  |
> | Commonsense   | 77.28 | 77.81 | 74.29  | 67.09 | 74.23 | 79.01  |
>
> ---
> **Q3. Concern on the Overstatement of the Magnitude Principle**
>
> We fully agree that the applicability and limitations of the magnitude principle deserve careful discussion and should not be overstated. Below, we clarify our intent and how we have already addressed some of these concerns:
>
> 1. **Primacy, Not Universality**.
>    1. As stated in our title, our central claim is that **magnitude plays a primary—not universal—role** in the success of various LoRA hyperparameter-tuning strategies. We do **not** argue that magnitude alone determines all outcomes, but rather that it provides a coherent and predictive lens across settings.
>    2. As indicated in lines 134–135, the magnitude principle is **not limited to initialization magnitudes** $\sigma_A^2$ and $\sigma_B^2$,  but also includes rank $r$, learning rate $\eta$, and gradient variance $\sigma_L^2$ (task-dependent).
>    3. Our LoRAM focuses on **the impact of initialization magnitude in PiSSA**, serving as a concrete evidence of the magnitude principle.
>
> 2. **Self-Consistency and Validation**.  We believe our statements are consistent and validated by extensive experiments and ablation studies, as recognized by other reviewers.
> 3. **Discussion of Limitations**.
>    We agree that further discussion will strengthen the paper. We have already addressed the following in the current version:
>    1. **Interaction with Rank Size**: As shown in lines 260–263, we empirically observe that the benefits of magnitude scaling **diminish and even reverse** at high ranks, aligning with our theoretical Proposition 2.
>    2. **Architectural and Task Diversity**: We deliberately evaluated our approach across a variety of architectures and tasks to avoid conclusions based on narrow settings and to support the generality of our findings.
>    3. **On Optimality**: We noted in lines 137–138 and 276–280 that our work does not aim to identify an optimal strategy. Rather, we aim to reveal the key factors that influence performance, leaving this challenging topic to future work.
>
> In our revision, we will further clarify these points and make the scope and limitations of our claims more prominent.
>
>
> ---
> **Q4. Moderate Improvement in Convergence and Performance Over Strong Baselines**
>
> We appreciate the reviewer’s observation. We would like to clarify the broader context and significance of this result:
>
> 1. **Simplicity Grounded in Theory as a Core Contributions**.
>    Our main contribution is not to claim dramatic acceleration in convergence, but to demonstrate that **comparable or better performance can be achieved with significantly lower complexity**. In this light, matching PiSSA’s convergence speed is itself a meaningful contribution as recognized by reviewers.
>
> 2. **Consistent Trend as Theoretical Validation**.
>    The fact that LoRAM tracks PiSSA’s performance is a validation of our central theoretical claim. Since LoRAM mimics the magnitude gain deterministically, the similarity in convergence trajectories supports our hypothesis and underscores the diminishing role of complex SVD-based initialization once magnitude is properly controlled.
> 3. **Potential for Further Gains**.
>    As shown in our ablation studies (Sec 4.3), LoRAM can be readily combined with other methods—such as RsLoRA—to achieve further improvements. Despite this, we would like to emphasize that **our work not only aims to push advancements**, but also to **identify and validate the fundamental mechanisms**.
>
> We believe the ability to **consistently deliver solid performance using a theoretically-grounded, explainable and resource-efficient approach** makes a significant contribution.
>
> ---
> [1] Lora: Low-rank adaptation of large language models. In ICLR, 2022.
>
> [2] Pissa: Principal singular values and singular vectors adaptation of large language models. In NeurIPS 2024.

---

> > ### Author Response · Authors · 2025-08-05
> > **Follow-up on Rebuttal Discussion**
> >
> > Dear Reviewer PbY3,
> >
> > Thank you again for your valuable time and thoughtful feedback on our submission.
> >
> > We would like to kindly follow up to check whether our response has addressed your concerns. If there are any remaining questions or suggestions, we would greatly appreciate the opportunity to clarify further.
> >
> > Your feedback is valuable to us and has been helpful in improving the quality of our work. Thank you again for your engagement.
> >
> > Best regards,
> >
> > The Authors of Submission 4604

---

> ### Comment · Reviewer_PbY3 · 2025-08-05
>
> It appears that PiSSA and MiLoRA may have different optimal hyperparameter configurations—particularly learning rates—which could explain some of the observed performance differences.
>
> From my own experience, MiLoRA tends to perform comparably to PiSSA or other recent baselines across a variety of tasks, and I have found learning rate to be a particularly sensitive factor influencing its performance. This discrepancy led me to question the reported underperformance of MiLoRA. Notably, it is still unclear from the appendix whether each baseline (including MiLoRA and PiSSA) was adequately tuned for the respective tasks and settings.
>
> Although, I appreciate the authors’ thoughtful rebuttal, the additional experiments, and the theoretical motivation provided. Considering these points, I raise my score to borderline accept, although my concerns regarding Q2 are not fully resolved.

---

> > ### Author Response · Authors · 2025-08-05
> > **Thanks for Your Response and Support.  Clarifications on Baseline Settings and Contextual Strengths of Methods**
> >
> > **Dear Reviewer PbY3,**
> >
> > Thank you for your reconsideration and for raising your rating to a positive score. We also appreciate your insightful comments and would like to take this valuable opportunity to provide a more detailed discussion.
> >
> > ---
> >
> > ### **Clarification on Hyperparameter Tuning for Baselines**
> >
> > We did not conduct per-method hyperparameter tuning during all the experiments. Instead, we deliberately applied a unified hyperparameter configuration across all methods to ensure a controlled and fair academic comparison. For example, for LLaMA-2 and LLaVA, we followed the official settings provided by the respective codebases. Importantly, since LoRAM and PiSSA share similar underlying principles, this makes a shared setup a reasonable and reliable choice for comparative evaluation.
> >
> > ---
> >
> > ### **Further Discussion on MiLoRA’s Performance**
> >
> > We are grateful that your observation about MiLoRA’s sensitivity to learning rate—motivating our additional experiments (as noted in Q2)—resonates strongly with our own experience and analysis. We agree that this remains an **open and interesting question** for explaining performance diversity, and we appreciate the opportunity for deeper discussion and consensus.
> >
> > Beyond the singular value amplification we have discussed, we would also like to provide an explanation for some high-level factors that influence a method's performance and usage.
> >
> > In practice, as you noted, we have also observed that the **optimal adaptation method often depends on the degree of domain shift and the quality of data**:
> >
> > * In many settings (especially industrial ones), today's foundation models—particularly closed-source models—already possess rich prior knowledge. Meanwhile, task-specific datasets may contain noisy or weakly aligned data, making generalization more important than aggressive adaptation.
> > * In such cases, full-parameter fine-tuning may overfit to noise data. MiLoRA, with its conservative update strategy, proves particularly well-suited for such scenarios. It strikes a balance between adaptation and generalization.
> >
> > This is also empirically supported by our results—for example, on MNLI and SST-2, where full-parameter tuning underperforms compared to LoRA, MiLoRA performs consistently well. To summarize, we consider this an important open question. Based on our findings and your suggestion:
> >
> > * Methods like **PiSSA**, which amplify update magnitude, tend to excel when **domain shift is large** and **rapid adaptation** is needed.
> > * In contrast, **MiLoRA**, with more conservative updates on minor components, is particularly effective in **low-shift or noisy-data scenarios**, where **generalization is essential**—a common situation in real-world applications.
> >
> > We will include the experiments using larger learning rates (from Q2) in the appendix and mention this point in our revised paper to fairly showcase MiLoRA’s strengths. These would reinforce our core message: **Different methods might shine under different conditions.** LoRAM **traces their shine**, and our theory offers a **promising lens to understand their shine**.
> >
> > We are sincerely grateful for your insights, which have significantly enriched this discussion.
> >
> > Best regards,
> >
> > The Authors of Submission 4604

---

### Decision · Program_Chairs · 2025-09-17

**Decision:**

Accept (spotlight)

**Comment:**

The paper introduces a novel method (LoRAM) and a new theoretical perspective that re-interprets spectral methods through magnitude scaling. The proposed LoRAM initialization achieves competitive or superior performance compared to leading LoRA schemes. The claims are supported by a wide range of experiments conducted across diverse tasks.

During the rebuttal process, most of the concerns over the results and motivation of the paper are adequately addressed by the authors, and two reviewers increased their final scores. However, some clarity issues on experiment results (e.g. MiLoRA) and figures still remain.

Considering the significant technical contribution and the consensus positive reviews, I would recommend the acceptance of this paper. I strongly suggest that the authors to continue to improve the paper's clarity on experimental results and overall representation quality as suggested by the reviewers for final publication.